## OPEN
# Event-triggered STED imaging

Jonatan Alvelid⬤, Martina Damenti, Chiara Sgattoni⬤ and Ilaria Testa⬤ ✉

**Monitoring the proteins and lipids that mediate all cellular processes requires imaging methods with increased spatial and temporal resolution. STED (stimulated emission depletion) nanoscopy enables fast imaging of nanoscale structures in living cells but is limited by photobleaching. Here, we present event-triggered STED, an automated multiscale method capable of rapidly initiating two-dimensional (2D) and 3D STED imaging after detecting cellular events such as protein recruitment, vesicle trafficking and second messengers activity using biosensors. STED is applied in the vicinity of detected events to maximize the temporal resolution. We imaged synaptic vesicle dynamics at up to 24 Hz, 40 ms after local calcium activity; endocytosis and exocytosis events at up to 11 Hz, 40 ms after local protein recruitment or pH changes; and the interaction between endosomal vesicles at up to 3 Hz, 70 ms after approaching one another. Event-triggered STED extends the capabilities of live nanoscale imaging, enabling novel biological observations in real time.**

STED (stimulated emission depletion) nanoscopy has been successfully used to image a variety of structures in both living cells and tissues, even dynamically[1]. The temporal resolution of STED nanoscopy usually depends on the size of the region of interest to be imaged, owing to its most common single point-scanning implementation. This means that the technique can achieve high frame rate imaging (1–30 Hz) in small regions of interest (1–5 μm²) (refs. [2–4]), but for larger fields of view, that is, up to $80 \times 80\,\mu m^2$, it takes on the order of minutes to acquire a single frame[5]. This illustrates the current trade-off between spatial and temporal resolution, in which fast dynamics inside cells can be followed only in sufficiently small areas that are often difficult to pinpoint due to the loss of the larger cellular context. Parallelized STED methods[6,7] have tried to overcome this trade-off by minimizing the number of scanning steps during imaging, but they are currently limited by the camera frame rate and depletion power. Moreover, although STED nanoscopy is capable of a high temporal resolution, it is also susceptible to photobleaching and photodamage, which limits the total number of recordable frames[8,9]. Techniques often called 'smart microscopy' attempt to provide solutions for gentler recordings by either adapting the illumination to the sample characteristics or by switching between microscopy modalities.

Sample-adaptive scanning approaches in conventional[10,11] and super-resolution microscopy[12,13] have helped to minimize the light dose and the recording time during imaging or to increase the image quality deep in tissues[14]. Multiscale approaches have focused so far on increasing the throughput of large-scale and relatively slow events such as cellular division[15–17] or on screening[18] with conventional methods, such as widefield or confocal imaging, or on super-resolution methods only after fixation. However, no sample-adaptive scanning approaches for live cell imaging have so far been triggered by subsecond real-time changes in the sample such as intensity spikes, local movements or morphological changes, and nor have they been automated to switch between distinct imaging modalities such as STED or other super-resolution approaches. As such, the increasingly important field of smart microscopy is still lacking a method that switches imaging scales and which incorporates nanoscopy methods that operate rapidly (on the order of tens of ms or seconds) after a stimulus.

It is currently difficult to observe cellular processes at high spatial resolution (~30 nm) efficiently and rapidly (up to tens of Hz)

in cells, either because they are hard to localize in large cellular volumes, they happen too fast, or because the number of frames before bleaching is not high enough. However, if the user knew where and when to image the cellular process of interest, the quality, throughput, speed, and length of the observation would increase substantially, enabling the dynamics of the process to be unraveled comprehensively.

Here, we present a novel sample-adaptive microscopy method called event-triggered STED (etSTED), which enables rapid two-dimensional (2D) and 3D STED nanoscopy acquisitions upon and at the site of automatic detection of subcellular events such as biosensing, local protein recruitment or vesicle trafficking in spatial proximity. It does so by combining fast (up to 20 Hz) widefield imaging, which facilitates the detection and localization of events, with STED imaging, for high-resolution acquisition at the site of a detected event. The STED imaging can be performed with lateral (2D STED) or axial (3D STED) super-resolution in one or multiple frames recorded sequentially. The maximal transition between widefield and STED imaging happens in a temporal window of 40 ms from an event taking place. To detect the events of interest, the method runs a real-time analysis pipeline on every recorded widefield frame. We took special care to develop an analysis pipeline fast enough to detect the events of interest with a minimum delay of 6 ms without compromising accuracy, that is, with minimal false-positive and false-negative events.

The generalized implementation of etSTED enables the investigation of a diverse combination of triggering events and fine subcellular structures. The analysis pipelines developed in our implementation detect events such as intensity spikes as in local calcium or pH sensing; slower rises in intensity upon protein recruitment as in dynamin-mediated endocytosis; or the spatial proximity of vesicles during subcellular trafficking. And etSTED imaging can instead be performed on various proteins including actin, tubulin and synaptotagmin, or lipids such as cholesterol and sphingolipids enriched in the plasma membranes and endosomal vesicles. With etSTED we can observe different types of vesicle and membrane fusion events comprehensively and with an unprecedented level of detail in both neurons and cancer cell lines. This was possible only with a detection speed and multiscale approach specific to this work, which complements previously developed, slower sample-adaptive imaging methods.

Department of Applied Physics and Science for Life Laboratory, KTH Royal Institute of Technology, Stockholm, Sweden. ✉e-mail: ilaria.testa@scilifelab.se

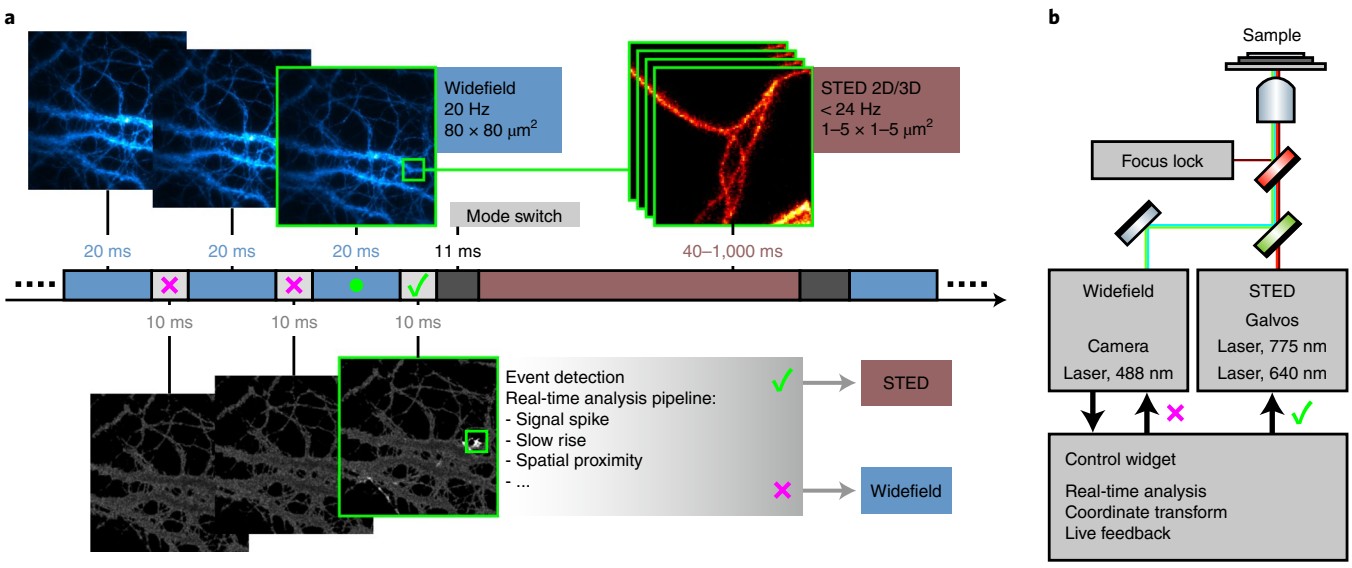

**Fig. 1 | Overview of event-triggered STED imaging. a**, Scheme of an etSTED experiment on a temporal axis with widefield calcium imaging of Oregon Green 488 BAPTA-1 in neurons in 20 ms (blue, top left images); corresponding analyzed images upon real-time application of an analysis pipeline in 10 ms (light gray, bottom images); switch of imaging modalities upon a detected event in 11 ms (dark gray); and a triggered locally scanned STED timelapse at the location of the detected event (red, top right image stack). Small green boxes indicate the chosen detected event that triggers STED imaging. **b**, Schematic diagram of the microscope set-up, combining STED and widefield imaging with real-time analysis, coordinate transformation and live visual feedback in a control widget implemented in the microscope control software.

## Results

**Overview.** Our automated etSTED acquisition scheme can trigger high-resolution STED imaging upon the detection of events in widefield images in the tens-of-milliseconds timescale (Fig. 1a). An $80 \times 80\,\mu m^2$ region of the sample is surveyed at any time in widefield imaging mode. These images are processed in real time with a rapid analysis pipeline, which returns a set of coordinates of any detected event. If an event, for example an intensity spike, is detected, the widefield imaging is stopped and the STED acquisition started in a small area around the detected coordinates with pre-determined image acquisition parameters. The size of the STED field of view is defined by the user, and in our applications it is always smaller than the widefield to achieve high temporal resolution (1–24 Hz). When the STED image or timelapse is acquired, comprehensive data regarding the event are saved. This includes a widefield timelapse leading up to the detected event, the scanned STED image or timelapse, and a log file summarizing the parameters and timings of the event detection and scanning. The saved auxiliary data are important to confirm the validity of the event in post-acquisition analysis. It also enables further quantification of the event both temporally and spatially, within the larger widefield field of view. The microscope (Fig. 1b) then returns to the previous settings and another continuous widefield recording is instantly initiated. The method can run indefinitely, and the focus lock in place in the microscope[5] maintains a stable sample throughout the experiment, which prevents axial drift and ensures that the widefield and STED images are being recorded in the same sample plane throughout a full experiment.

In this work three different analysis pipelines for detecting four types of events have been developed (Extended Data Fig. 1).

The rapid_signal_spikes analysis pipeline has been developed to detect local calcium intensity spikes in hippocampal neurons and HeLa cells labeled with the calcium-sensing probe BAPTA-1 (Extended Data Fig. 1a, Supplementary Fig. 1 and Supplementary Note 1). However, intensity spikes are not unique for calcium sensing but can also be generated by commonly used pH sensors

such as pHluorin. The appearance of spots in CD63-pHluorin-expressing HeLa cells during exocytosis has also been observed and used as triggering events (Extended Data Fig. 1b and Supplementary Note 1).

The dynamin_rise analysis pipeline can instead detect a slowly rising fluorescence signal, which can be generated by the local recruitment of specific proteins in a cellular locus. As an example we detected the recruitment of dynamin1 labeled with enhanced green fluorescent protein (EGFP) to the vicinity of the plasma membrane during endocytosis (Extended Data Fig. 1c and Supplementary Note 2).

The third analysis pipeline is named vesicle_proximity and it has been developed to track the position of endosomal vesicles and can detect when they are in fine spatial proximity. The analysis is demonstrated for CD63-EGFP-positive endosomal vesicles in hippocampal neurons (Extended Data Fig. 1d and Supplementary Note 3).

These three different ways to detect cellular events proves the versatility of the method and demonstrates its ability to be used with any type of event-detection pipeline.

The custom-built set-up (Fig. 1b and Supplementary Fig. 2) combines STED with widefield imaging by spectrally separating the two techniques and is controlled with the open-source software ImSwitch[19]. The control software integrates all hardware, allowing fast control of the components of both widefield and STED acquisition modes. This ultimately enables the event-triggered method to reach rapid (tens of milliseconds) feedback between the two imaging modalities and their hardware components: lasers, scanners, cameras and acousto-optic modulators and tunable filters. The etSTED method is fully controlled via a widget in ImSwitch, also released as a standalone widget to facilitate its implementation in other microscope control software (Supplementary Note 4), and the method is explained in detail in Supplementary Note 5. Furthermore, we used a third-order polynomial coordinate transformation[20] between the widefield space and scanning space to validate the accurate transformations across the field of view (Extended Data Fig. 2a). The

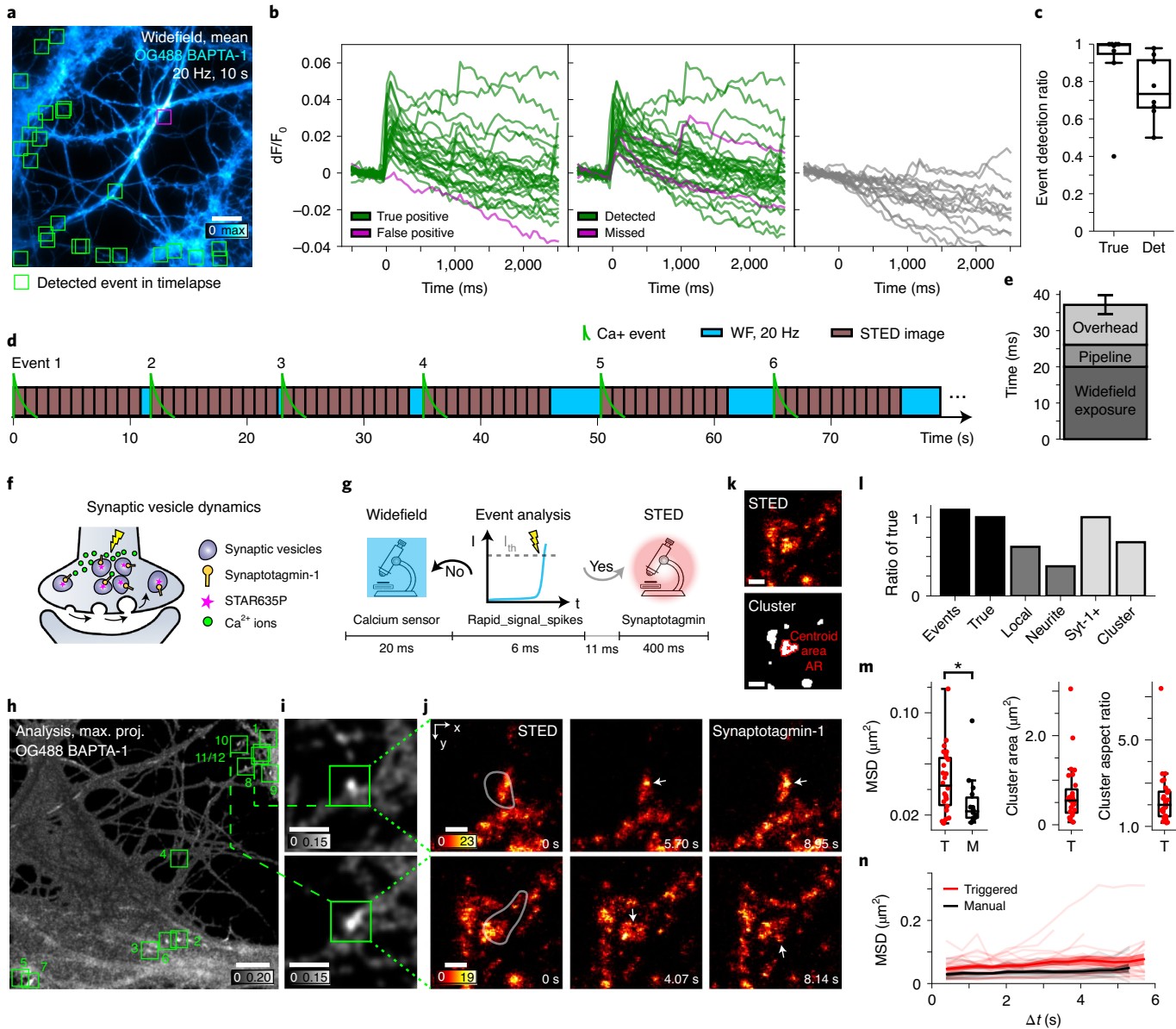

**Fig. 2 | Neuronal functional and structural imaging with etSTED. a**, Mean Oregon Green 488 BAPTA-1 widefield image from a 10 s timelapse. Boxes indicate detected events (green, true; magenta, false). $n = 25$ events. **b**, Extracted calcium curves at detected events (left), manually annotated events (center), and random positions inside the neuron (right). Fluorescence intensity $dF/F_0 = (F(t) - F(t_0))/F(t_0)$. **c**, Characterization of true-positive event detection ratio (True) and detected annotated event ratio (Det). $n = 8$ cells. **d**, Representative etSTED experiment. **e**, etSTED performance with the analysis pipeline rapid_signal_spikes. $n = 90$ events, $n = 11$ cells. **f**, Synaptic vesicle dynamics upon calcium signaling. **g**, Experiment timeline for one widefield frame. **h–j**, etSTED experiment with calcium signal-triggered STED imaging of synaptotagmin-1_STAR635P. **h**, Maximum-projected analysis image. Green boxes indicate detected events. **i**, Zoom-ins of the ratiometric image at the location of two detected events. Green boxes indicate the STED scan area. **j**, 2.5 Hz etSTED timelapses. White outlines indicate the detected local calcium activity area. Arrows indicate points of structural reorganization. **k**, Synaptic vesicle cluster analysis. AR, aspect ratio. **l**, Event detection ratios, compared with the number of true events, for number of events, local or neurite-wide calcium events, and timelapses with synaptotagmin-1-positive vesicles (Syt-1+) and vesicle clusters. $n = 186$ events, $n = 17$ cells. **m,n**, Analysis of synaptic vesicle clusters in calcium-triggered etSTED (red) and manual STED (black) timelapses: MSD with $\Delta t = 1$ frame (**m**, left), cluster area (**m**, center) and aspect ratio (**m**, right), and time-dependent MSD (**n**). **m**, The statistical test used is a two-sample two-sided Kolmogorov–Smirnov test: $P = 0.018$, test statistic $= 0.48$. **n**, Individual clusters (semi-transparent curves), mean curves (solid lines), and 83% confidence intervals (shaded areas). red: $n = 26$ clusters, $n = 10$ cells; black: $n = 14$ clusters, $n = 14$ cells. Box plots (**c,m**) show the 25–75% interquartile range, with the middle line representing the mean, and the whiskers reaching 1.5-fold the first and third quartiles. Bar plots (**e,l**) show the mean, and the whiskers reach ±1 s.d. Scale bars, 10 μm (**a,h**), 3 μm (**i**) and 500 nm (**j, k**).

transformation is calibrated by detecting fluorescent beads in the two imaging modalities and fitting the coefficients in the polynomial transformation. The accuracy was analyzed by transforming the widefield coordinates of all detected beads into scanning space

and comparing the transformed coordinates with those of the detected beads in the scanned image. The distance between coordinate pairs, transformed and detected, was calculated to generate a map of the accuracy, which shows no spatial correlation and a mean

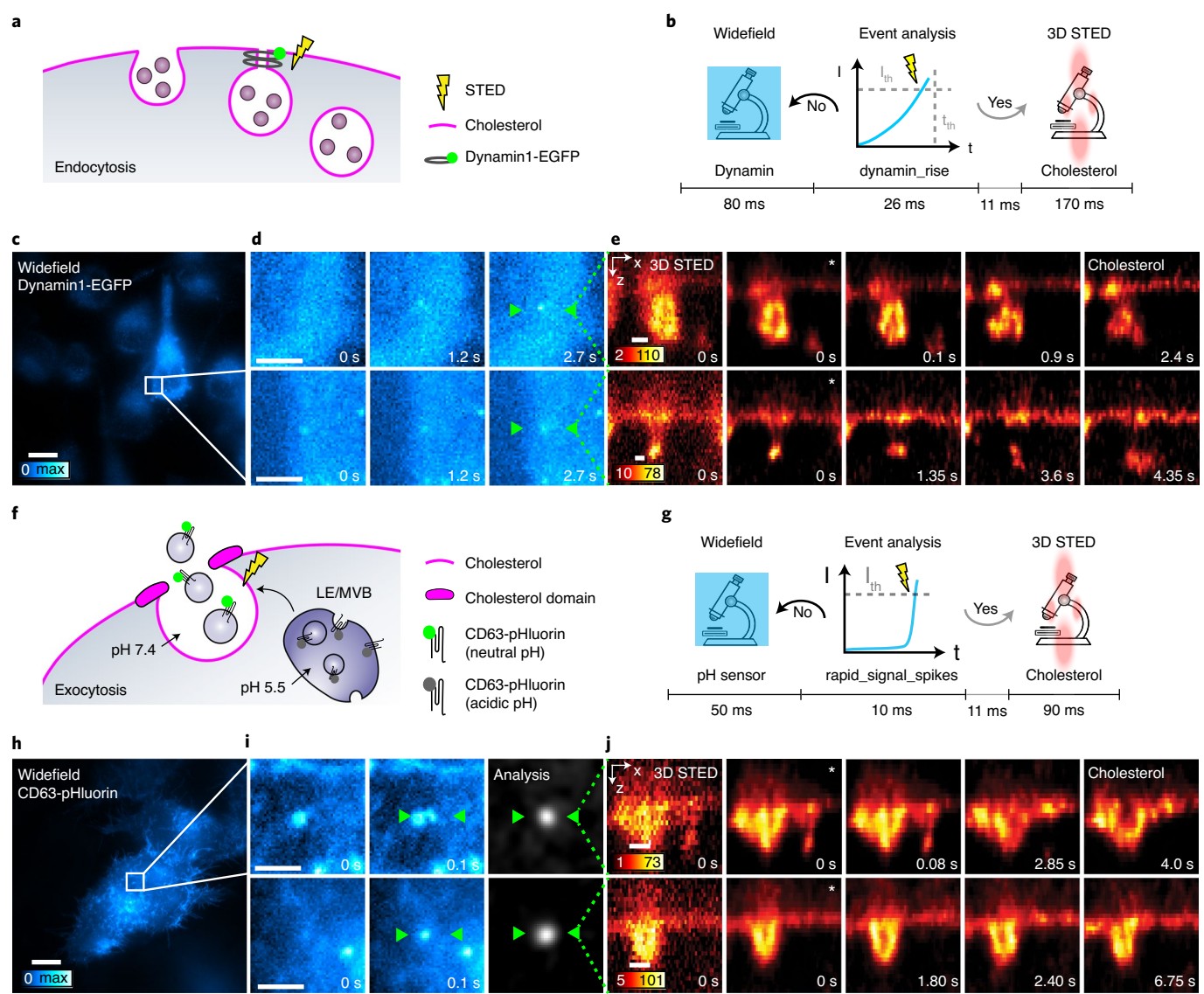

**Fig. 3 | Investigation of endocytosis and exocytosis with etSTED.** This shows an etSTED experiment in HeLa cells expressing Dynamin1-EGFP or CD63-pHluorin and using the dynamin_rise or rapid_signal_spikes analysis pipeline. **a**, Schematic diagram of the dynamin-mediated endocytosis process of interest, with the increase in intensity due to accumulation of dynamin1 at the endocytic site. **b**, Schematic diagram of the experiment timeline of one widefield frame. $I_{thr}$, intensity threshold; $t_{thr}$, time threshold. **c**, Example of a triggering widefield frame in an etSTED experiment with HeLa cells expressing Dynamin1-EGFP. **d**, Two representative events shown in widefield zooms with three frames leading up to the event (right frame). **e**, Triggered 5.9 Hz 3D STED timelapses of plasma membrane dynamics where cholesterol (cholesterol-Abberior STAR RED) is labeled. $n = 53$ events, $n = 22$ cells. **f**, Schematic diagram of an exocytosis event of interest, with the increase in fluorescence intensity due to unquenching of pHluorin upon pH neutralization of late endosomes (LEs) and multivesicular bodies (MVBs). **g**, Schematic diagram of the experiment timeline of one widefield frame. **h**, Example of a triggering widefield image in etSTED experiment with HeLa cells expressing CD63-pHluorin. **i**, Two representative events shown in widefield zooms (cyan, left) and analysis ratiometric image zooms (gray, right) with the last two frames before the event. **j**, Triggered 11 Hz 3D STED timelapses of plasma membrane dynamics and accumulation where cholesterol (cholesterol-Abberior STAR RED) is labeled. $n = 232$ events, $N = 29$ cells. Asterisks (*) indicate deconvolved frames, and apply to all following frames in the same timelapse. Scale bars, 10 μm (**c,h**), 2 μm (**d,i**), 250 nm (**e,j**).

transformation error of 54 nm across the field of view, which can be compared to the widefield pixel size of 100 nm. To confirm the spatial resolution and quality in the triggered STED images across the field of view, we performed imaging of microtubules at local calcium activity sites using the rapid_signal_spikes analysis pipeline (Extended Data Fig. 2b–e and Supplementary Note 6). The width of the microtubules was measured, resulting in a mean width of $46 \pm 11$ nm. Considering the convolution process between the point spread function (PSF) and structure during the imaging, as well as

the physical size of the microtubules and labeling molecules, the spatial resolution is estimated to be ~30 nm.

**STED imaging triggered by calcium activity.** We applied the rapid_signal_spikes analysis pipeline with etSTED imaging to detect local calcium activity events and study the rearrangement of synaptic proteins at high spatiotemporal resolution during calcium activity in hippocampal neurons (Fig. 2). The pipeline extracts coordinates relating to events of rapid intensity increases in the

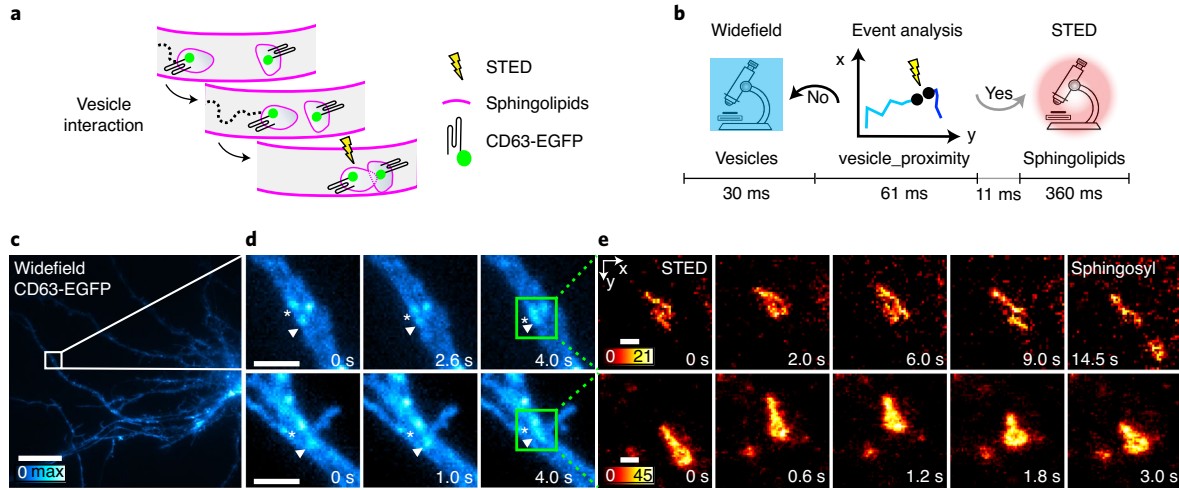

**Fig. 4 | Investigation of endosomal vesicle interaction with etSTED.** This shows an etSTED experiment in hippocampal neurons expressing CD63-EGFP and using the vesicle_proximity analysis pipeline. **a**, Schematic diagram of an endosomal vesicle interaction process, with the increasing proximity due to one labeled vesicle moving towards another stationary vesicle as labeled with CD63-EGFP. **b**, Schematic diagram of the experiment timeline of one widefield frame. **c**, Example of a triggering widefield frame. **d**, Two representative events shown in widefield zooms with three frames leading up to the event (right frame). The green box indicates the area of the STED scan. **e**, Triggered 2.8 Hz STED timelapses of endosomal vesicle dynamics where sphingosyl PE (sphingosyl-PE_Abberior STAR RED) is labeled. $n = 123$ events, $n = 23$ cells. Scale bars, 10 μm (**c**), 2 μm (**d**), 250 nm (**e**). The vesicles involved in the event are marked by symbols (asterisks and arrowhead).

sample by peak detection in a calculated map of ratiometric intensity changes, comparing the current and previous widefield frames (Extended Data Fig. 1a, Supplementary Fig. 1 and Supplementary Note 1). The authenticity and efficiency of the detection of calcium spikes in neurons was investigated (Fig. 2a–c). By applying the analysis pipeline after acquisition on an uninterrupted widefield timelapse recording (Fig. 2a) we could extract calcium intensity traces at the sites of detected events also after the detection (Fig. 2b, left). We can confirm that the detected events are, on average, true calcium spike events in 91% (0.91 ± 0.19) of the cases (Fig. 2c), as shown by the characteristic initial spike and decay across hundreds of milliseconds. Moreover, in more than half of the analyzed timelapses, 100% of the detected events were true. A few of the detected events in the timelapses do not show a real spike at $t = 0$, and through manual annotation of the timelapses it is confirmed that these are false detections, and in most cases they are due to the rapid movement of small filopodia-like structures in neuronal cells. It is important to note that these events can be sorted out in post-acquisition analysis of the saved data by inspecting the calcium trace up to the point of event detection (Supplementary Fig. 3 and Supplementary Note 7). The detected movement suggests another potential application of the method whereby detection of cellular movement can be used to trigger STED imaging. Furthermore, by manually annotating calcium events in the timelapse recordings we can see which of the events are detected with the analysis pipeline (Fig. 2b, middle). On average, 76% (0.76 ± 0.16) of the events in the neurons are detected by the analysis pipeline (Fig. 2c), and only a few are missed, with generally a lower ratiometric peak value that is below the applied ratiometric threshold. This means that the true-positive event detection ratio (0.91) is higher than the detected events ratio (0.76), which is due to the pipeline being optimized for accuracy to minimize false detections, but regardless detects most of the events. Last, to further confirm the detected events as calcium spikes, we extract calcium traces from random positions inside the neurons in the very same timelapses (Fig. 2b, right), which show expected flat calcium signal profiles.

A typical etSTED experiment in neurons can detect several calcium spikes sequentially, and dynamically switch between the imaging techniques. In a representative experiment, the duration of widefield imaging between the end of the previous STED timelapse and the next detected event varied between 0 and 5 s (Fig. 2d). The analysis of the temporal performance shows that, for a widefield imaging experiment with an exposure time of 20 ms and 800 × 800 pixels, the analysis pipeline takes 6.1 ± 0.4 ms to run (Fig. 2e). The analysis pipeline is implemented to run on a GPU (graphics processing unit), and thus the speed is improved up to fivefold over CPU (central processing unit) implementations (Supplementary Fig. 4 and Supplementary Note 8). The time for the coordinate transform is <1 ms. The overhead time, which includes toggling of hardware, scanning curve calculation and scan initiation, is 11.0 ± 2.9 ms. Overall, the total triggering process from widefield exposure initiation to STED scan start in the case of a detected event, takes 37.1 ± 3.4 ms. This means that the maximum time lag between an event taking place and the STED scan starting at the site of that event is ≤40 ms (Fig. 2e).

The analysis pipeline was applied to trigger STED imaging of actin (SiR-actin) and microtubules (SiR-tubulin), as timelapses over 11 frames at 1.0 Hz in a 5 × 5 μm² field of view (Extended Data Fig. 3, Supplementary Figs. 5,6 and Supplementary Note 6). Frames from two triggered STED timelapses show nanoscale actin structures in dendritic spines as well as the membrane periodic skeleton (Extended Data Fig. 3). The experiment lasted 2 min 46 s, from the detection of event 1 to the end of the STED timelapse following event 13.

In the presynaptic active zone the voltage-dependent, local and transient increase in intracellular calcium leads to vesicle release from the readily releasable synaptic pool[21,22]. The protein synaptotagmin-1 (Syt-1) is the calcium sensor that triggers vesicle release. The replenishment of the recycling pool is then mediated by further endocytosis, which occurs from tens of milliseconds up to 20–30 s from the event[23]. Although the dynamics of synaptic vesicles upon electrical and chemical stimulation have been studied[24,25], their concurrent behavior upon basal activity is still unexplored due

to technical limitations in accessing such temporal window rapidly enough and with sufficient spatiotemporal resolution.

We use etSTED to image the reorganization of synaptic vesicles in active synapses during calcium sensing (Fig. 2f,g, Extended Data Fig. 4 and Supplementary Note 6). Antibodies against the intraluminal domain of Syt-1 are pre-mixed with secondary nanobodies conjugated to Abberior STAR635P and incubated with live neurons. Upon fusion of the vesicles with the plasma membrane, the Syt-1 intraluminal domain is exposed to the extracellular milieu, enabling binding of the nano-antibody[26]. When the synaptic vesicles are further recycled, the internalization of the labeled molecules enables imaging of the vesicle pools inside the presynaptic active zone (Fig. 2f). Areas of activity of both local calcium events, approximately <5 μm² in size, as well as spread-out neurite-wide events are shown in the maximum projection of the analyzed ratiometric images (Fig. 2h). The etSTED experiment had 12 detected calcium events and lasted 4 min 24 s (Fig. 2h, green boxes). We show ratiometric widefield images from the analysis pipeline (Fig. 2i) of two detected local events. The STED timelapses following the events (Fig. 2j and Extended Data Fig. 4), consisting of 31 frames taken at 2.5 Hz in a 3×3 μm² region, show dynamic activity of the synaptic vesicles. In total we imaged 186 events in 17 areas of cells, and 91% of the detected events were true calcium events (Fig. 2l), and of those, 62% were local calcium spikes and 38% were neurite-wide or larger events (Fig. 2l). Furthermore, every true calcium spike event had Syt-1-positive vesicles in the vicinity, and 68% of them corresponded to clusters of synaptic vesicles (Fig. 2l). Rearrangements of the shape, density and connectivity of the densely packed synaptic vesicles such as disassembly of the clusters and individual vesicle dynamics are all captured in the STED timelapse recordings (Fig. 2j). We quantified the mean square displacement (MSD), area and aspect ratio of the clusters in the active synapses (Fig. 2k). The area of the clusters was $0.68 \pm 0.63$ μm² (Fig. 2m, middle) and the aspect ratio was $2.3 \pm 1.2$ (Fig. 2m, right). We also imaged similar vesicle clusters with manual STED imaging (Extended Data Fig. 5 and Supplementary Note 9) and compared the MSD over the timelapses in the two cases (Fig. 2m,n). With calcium-triggered etSTED timelapse imaging we were able to record faster vesicle cluster dynamics that occur within calcium oscillations. The MSD with a $\Delta t = 1$ frame (0.41 s) was $0.046 \pm 0.025$ μm² in the calcium-triggered timelapses and $0.030 \pm 0.020$ μm² in the manual timelapses (two-sample two-sided Kolmogorov–Smirnov test: $P = 0.018$, test statistic, 0.48) (Fig. 2m, left). Furthermore, the time-dependent MSD shows a clear separation of the two curves at all values of $\Delta t$ (Fig. 2n). Synaptic vesicle dynamics upon calcium activity is, therefore, faster than those in averaged synapses without calcium synchronization.

While the above STED imaging was performed at frame rates up to 2.5 Hz, further decreasing of the size of the imaged STED region around the detected event of interest enables us to increase the frame rate. The dynamics of the individual synaptic vesicles as well as the vesicle clusters were further captured at 24 Hz STED imaging in regions of 1×1 μm² (Extended Data Fig. 6), proving the capability of the method to reach video-rate imaging speeds at the events of interest. Altogether, etSTED enables the monitoring of calcium-signaling-triggered dynamics of synaptic vesicles in basal conditions. Based on the temporal window of the vesicle displacement, it is likely that we are catching their endocytosis to replenish the recycling pool[27,28].

**pH sensing- and dynamin-triggered 3D STED imaging of cytosis.** We also used etSTED to image endocytosis in HeLa cells (Fig. 3a–e). In the cellular plasma membrane (PM), macromolecules are rapidly internalized via endocytosis from the extracellular compartment. The GTPase protein dynamin1[29] is crucial in several mechanisms of endocytosis by accumulating at and wrapping around

the neck of budding vesicles, promoting membrane fission and internalization[30]. In these events, cholesterol plays a central role in promoting PM curvature[31]. To detect dynamin1-dependent endocytosis events, we expressed dynamin1-EGFP in HeLa cells and added cholesterol-Abberior STAR RED in solution (Fig. 3a). The slow rise (~seconds) in signal due to the accumulation of dynamin1 at the endocytic site was detected in the widefield imaging using the dynamin_rise analysis pipeline (Supplementary Note 2) and was used as the triggering event (Fig. 3b–d). This enabled us, within 120 ms of a detected event, to apply 3D STED imaging in x–z cross-sections at the sites of detected events (Fig. 3d,e). The 3D STED imaging was performed at 6–9 Hz to capture the dynamics of the PM enriched with cholesterol, which enabled the observation of the budding and fission of the endocytic vesicle (Fig. 3e and Extended Data Fig. 7). We could follow the events continuously for 5–10 s. In some cases we observed multilobed vesicles formed during the endocytosis process[32].

Once internalized, vesicles are transported along the endo-lysosomal pathway[33,34]. In the endo-lysosomal pathway, the role of multivesicular bodies (MVBs) is crucial for vesicular structures, which contain small intraluminal vesicles. They can be either sorted to lysosomes for degradation or fused to the PM for exocytosis of exosomes[35]. Intraluminal vesicles are highly enriched in the protein tetraspanin CD63, cholesterol and sphingolipids[36]. In particular, MVB–PM fusion is stimulated by cholesterol nanodomains promoting a negative curvature of the membrane following the fusion event[37].

To image the dynamics of plasma membrane cholesterol in the rare and delocalized MVB–PM fusion events, we expressed CD63-pHluorin in HeLa cells and added cholesterol-Abberior STAR RED in solution[38] (Fig. 3f–j). We used the rapid_signal_spikes analysis pipeline to detect the unquenching of the pH-sensitive GFP following pH neutralization upon MVB–PM fusion, which results in a rapid rise in fluorescence intensity and which was used as the triggering event (Fig. 3g–i). This allowed us to rapidly visualize plasma membrane cholesterol dynamics within 70 ms after the event taking place with 3D STED imaging at 7–11 Hz. Here, the omega shape of the PM following MVB fusion can be followed over time (Fig. 3j). We also found cases of bright accumulation of cholesterol (Extended Data Fig. 8). We propose that these cholesterol hot spots are part of a ring positioning the exocytic site before cholesterol diffusion within the omega shape or adjacent exocytosis events.

**STED imaging triggered by endosomal trafficking.** In between endocytosis and exocytosis, vesicle interaction and fusion are required to mediate endo-lysosomal pathway maturation and to allocate macromolecules that need to be degraded or recycled[39–41]. The allocation of enzymes and lipids is crucial, of which, cholesterol and sphingolipids are remarkably important for highly polarized cells such as neurons. Here, we used widefield imaging and the analysis pipeline vesicle_proximity (Supplementary Note 3) to track CD63-EGFP-positive vesicles in neurons (Fig. 4). The pipeline localizes and tracks all vesicles from frame to frame of the widefield imaging, and keeps a number of previous timepoints in memory. At every frame it additionally checks for disappearing tracks, and when it finds them it performs multiple Boolean checks. It looks for another track in the close vicinity, if the track disappeared a number of frames ago, if the tracks have been present in the last frames, and if at least one of the tracks has moved a certain distance in the last frames. If all checks return true, it signifies that there are two tracked vesicles that have approached each other and remained in close proximity, inside the spatial resolution of the widefield imaging, for at least hundreds of ms (Fig. 4a,c,d), and STED timelapse imaging is triggered (Fig. 4b). This enabled us to verify or dismiss potential interaction events of sphingolipids and cholesterol-positive endosomal vesicles in triggered STED

timelapses (Fig. 4e and Extended Data Fig. 9). For example, many vesicles seemingly undergoing fusion as seen in the widefield imaging are resolved as separate vesicles in the STED images. With our STED resolution we can additionally often observe tubular-shaped lipid structures[42] (Fig. 4e, top). Although the analysis pipeline is here used to detect events of potential interaction of CD63-positive vesicles, it is generalizable to widefield imaging of other endosomal proteins or lipids, and hence can be applied directly in many other vesicle-trafficking studies.

## Discussion

In this work we outline an event-triggered method, etSTED, which enables super-resolved STED imaging within 40 ms from a detected event taking place. Directing STED imaging in selected small regions around the detected events and within a specific time window does not only minimize the overall cell stress and photodamage but also enables a significant increase in the temporal resolution as compared with conventional STED imaging. The selected regions of interest can be imaged 100–5,000-fold faster as compared with the manual acquisition of STED timelapses in the whole investigated sample region (frame periods with the same acquisition settings: 218.1 s ($80 \times 80 \, \mu m^2$), 336 ms ($3 \times 3 \, \mu m^2$), 41.3 ms ($1 \times 1 \, \mu m^2$)). Although the STED frame acquisition time is currently the rate-limiting step, we could imagine future developments in which an even smaller and event-adapted region is imaged, which would further increase the possible STED timelapse frequency beyond the 20–30 Hz presented here. With regards to the technical implementation, the processing time of the analysis pipelines as well as the total time from the event to the STED imaging could be further minimized with an increase in computational power and faster camera technology.

Moreover, the sample-adaptive acquisition method enables experiments not previously feasible due to human reaction times, in which, for example, the dynamics and structure of synaptic vesicle pools can be directly correlated to the presence or lack of local calcium oscillations during basal neuronal activity. The possibilities to generalize and extend the method are endless: the super-resolved imaging can be extended, for example, to RESOLFT[43] and MINFLUX[44] microscopy, depending on the labeling strategy, duration of timelapse imaging and spatial resolution desired. Although our rapid_signal_spikes pipeline has been developed for detecting rapid, frame-to-frame intensity variations in widefield images (~50 ms in our applications) and has been proven to be efficient at localizing both calcium activity and local pH changes, the detection of other signaling and trafficking events might require tailored analysis pipelines. We provide two examples of such pipelines, named dynamin_rise and vesicle_proximity, which detect more slowly rising signals (~1 s) and the increasing proximity of endosomal vesicles, respectively. Together this shows the potential of using diverse types of cellular dynamic events as triggers. The current implementation in the control software (Supplementary Note 4) enables an easy exchange of analysis pipelines, such that each can be optimized to detect the event of interest with the desired balance of accuracy and speed. To further promote the implementation of our rapid multiscale method in other microscopes we provide a generalized and stand-alone widget (Supplementary Note 4), which is available for direct implementation in other Python-based control software. For application in control software developed in other programming languages, a detailed template of implementation is also provided (Supplementary Note 5).

Overall, the etSTED method bridges the gap between two types of imaging: gentler and photon-efficient large-scale imaging on the tens-of-milliseconds timescale, and small-scale super-resolution imaging at high temporal resolution. Given that this is done in real time, it fosters new applications in the functional and structural imaging of dynamic subcellular processes.

## Online content

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

## Methods

**Microscopy set-up.** All images were acquired on a custom-built STED and widefield set-up, based on a STED set-up previously described[5]. A complete schematic and list of components is given in Supplementary Fig. 2 and Supplementary Note 10. For STED, excitation of red-shifted dyes was done with a pulsed diode laser at 640 nm (pulse width 60 ps, LDH-D-C-640, PicoQuant). Depletion was done with a pulsed 775 nm laser beam (pulse width 530 ps, KATANA 08 HP, OneFive). Control of excitation and depletion laser illumination is done using an acousto-optic tunable filter (AOTFnC-400.650-TN + MPDS4C-B66–22–74.156, AA Opto Electronic) and an acousto-optic modulator (MT110-B50A1.5-IR-Hk + MDS1C-B65–34–85.135-RS, AA Opto Electronic). Orthogonal polarizations of the depletion beam are shaped using vortex and top-hat phase masks on a polarization-sensitive spatial light modulator (LCOM-SLM X10468–02, Hamamatsu), and the polarization directions are temporally delayed to avoid interference. Wave plates are used to create circular polarization necessary for optimal depletion focus formation. Galvanometer mirrors are used for scanning (6215H + 671215HHJ-1, Cambridge Technology) in a scanning system to enable constant resolution across an $80 \times 80\,\mu m^2$ field of view, as previously described[5]. Fluorescence is decoupled with a dichroic mirror. A bandpass filter (GT670/40m, Chroma) and a notch filter (NF03–785E-25, Semrock) are used before detection with an avalanche photodiode (SPCM-AQRH-13-TR, Excelitas Technologies). For the widefield set-up, excitation of blue-shifted fluorophores was done with a modulated 488 nm continuous-wave diode laser (06-MLD 488 nm, Cobolt). Detection of widefield images was done through a bandpass filter (FF01–540/80–25, Semrock) and a notch filter (ZET785NF, Chroma) with an sCMOS (scientific complementary metal oxide semiconductor) camera (Orca Flash 4.0, Hamamatsu). The widefield path is coupled into the beam path with a dichroic mirror after the scanning system. The general set-up uses a ×100/1.40 oil immersion objective (HC PL APO ×100/1.40 Oil STED White, Leica) and a microscope stand (DMi8, Leica). The system uses a mechanical stage for lateral sample movement (SCAN IM 130×85, Märzhäuser) and a piezostage for axial sample movement (LT-Z-100, Piezoconcept).

**Microscope control and computer.** Microscope hardware control is mainly performed through a National Instruments data acquisition (NI-DAQ) acquisition board (PCIe-6353, National Instruments). Hardware is controlled using microscope control software ImSwitch[19] written in Python. Control of the etSTED method is performed using a custom-written widget and controller in ImSwitch, available at GitHub (https://github.com/kasasxav/ImSwitch and https://github.com/jonatanalvelid/ImSwitch-etSTED), which controls lasers, image acquisition, and runs real-time analysis pipelines with customizable parameters. Instructions on how to run etSTED imaging can be found in the GitHub repository of the standalone widget (https://github.com/jonatanalvelid/etSTED-widget), while instructions on how to run ImSwitch can be found in the repository on GitHub and corresponding documentation (https://imswitch.readthedocs.io).

A focus lock controlled with ImSwitch combining an infrared laser (CP980S, Thorlabs), a CMOS camera (DMK 33UP1300, The Imaging Source) and the z-piezostage through a feedback loop, as previously described[5], is used. It enables experiments to run stably for time periods longer than hours.

The microscope control computer contains a Ryzen 7 3700X CPU (AMD) and a GeForce RTX 3060 Ti GPU (ASUS).

**etSTED widget and real-time image analysis pipelines.** The etSTED widget is a software module built in a generalized way to enable free choice of the lasers and detectors controlled with ImSwitch. For a thorough description of how to perform etSTED experiments with the control software, see the ImSwitch-etSTED GitHub repository readme file. The widget enables any arbitrary analysis pipeline to be used, and it additionally enables calibration of the imaging space coordinate transform.

The coordinate transformation calibration is performed in a help widget into which a pre-acquired widefield image and scanning image of the same sample area can be loaded. Manual annotation of the same sample points in the images prepares fixed points for the coordinate transform calibration. A general third-order polynomial transformation with Levenberg–Marquart optimization is used, enabling it to be compatible and precise regardless of any aberrations and distortions that may be present in the optical paths.

Afterwards, an image analysis pipeline can be loaded, and all editable parameters of the pipeline will be loaded into the GUI (graphical user interface). Last, to prepare for an etSTED experiment to be run, a binary mask of the sample may be recorded. It is performed with a separate functionality in which 10 frames are recorded and averaged, and a global thresholding is performed with a user-provided intensity threshold. The etSTED experiment is then started, with various options present as choices in the GUI: perform it as an endless loop or a single trigger; in visualization mode with real-time visualization of the preprocessed images to optimize analysis pipeline parameters; or in validation mode without triggering scans. While running etSTED experiments, detected event coordinates will be overlaid on the displayed widefield frames. The triggered imaging will be performed using pre-determined scanning parameters, including the choice of which lasers to use.

Real-time analysis pipelines are provided as standalone python functions and must take the current widefield image, the previous widefield images and a binary mask of the considered region as input, and return a list of detected event coordinates in the widefield space. The real-time analysis pipeline additionally takes and returns a variable with any user-defined information from the analysis runs of the previous frames, for example for pipelines requiring tracking. It can additionally take any numerical parameters as input, for example various thresholds. The analysis pipelines developed in this work are provided and explained in more detail below, in Supplementary Notes 1–3, as well as in the GitHub repository of the standalone widget (https://github.com/jonatanalvelid/etSTED-widget). Analysis pipeline parameter values used for each experiment are listed in Supplementary Table 1.

The etSTED widget and analysis pipelines in the control software use the Python packages numpy[45], scipy[46], cupy, opencv, trackpy[47], pandas[48], napari (www.napari.org) and pyqtgraph.

**Rapid signal spikes image analysis pipeline.** One analysis pipeline used throughout this work, rapid_signal_spikes, was developed to detect calcium events with Oregon Green 488 BAPTA-1 labeling in hippocampal neurons as well as CD63-pHluorin signal peaks in HeLa cells. It is shown schematically in Supplementary Fig. 1, further described in Supplementary Note 1, and available in the GitHub repository of the standalone widget (https://github.com/jonatanalvelid/etSTED-widget/blob/main/analysis_pipelines/rapid_signal_spikes.py). Although it is optimized for the above-mentioned biosensors, it is also likely to perform well with similar, fast fluorescence sensors after pipeline parameter tweaking. The pipeline consists of pre-processing that uses the current frame, the previous frame, and a binary region-of-interest mask; and peak detection. The pre-processing transforms the current image into a smoothed map comparing the intensity in each pixel between the current and previous image. The peak detection compares the image with a maximum-filtered version of itself, finding local maxima as the coordinates where the two are equal. To avoid detection of noise fluctuations, the coordinate intensities are thresholded. Finally, the ratiometrically brightest peak is used throughout this work as the coordinate where the triggered STED imaging takes place. Most of the analysis pipeline runs on the GPU, significantly decreasing the runtime as compared with running it on the CPU. Altogether, the analysis pipeline runs in 6 ms for the $800 \times 800$ pixels widefield images used in this work.

**Rising signal image analysis pipeline.** A second analysis pipeline used in the work is dynamin_rise, which detects slowly rising signal peaks occurring over multiple frames. We show this pipeline applied to dynamin1-GFP rising-signal-peak events in HeLa cells. It is further described in Supplementary Note 2 and is available in the GitHub repository of the standalone widget (https://github.com/jonatanalvelid/etSTED-widget/blob/main/analysis_pipelines/dynamin_rise.py). The pipeline performs pre-processing with smoothing and background reduction. Then, a peak detection similar to that of rapid_signal_spikes is performed, and high and low thresholds are applied to avoid detecting noise and large clusters. Following this, the intensity in a small area around each peak is extracted. The peak positions and intensities are stored and form an extra-info parameter, and the pipeline links tracks from the peak positions. These tracks are analyzed to identify when a certain track first appears and how the intensity of that peak develops over time. An event is triggered after the appearance of a trace that stays detected for N frames (user-definable) and has an intensity that ratiometrically increases above a certain threshold ratio over those frames. The last coordinate of that trace is the event coordinate. Again, with substantial parts of the pipeline running on the GPU, it runs in 20–60 ms for an $800 \times 800$ pixels widefield image, depending on the number of tracks followed. The pipeline is likely to work after parameter tweaking for other similar event detection tasks in which local intensities are increasing over multiple frames.

**Vesicle proximity image analysis pipeline.** The third analysis pipeline used in this work, vesicle_proximity, detects incipient proximity-of-vesicles events, such as two endosomes approaching one another. We apply this pipeline to look at events in which multiple CD63-GFP-positive vesicles approach one another, signifying interaction. It is further described in Supplementary Note 3 and is available in the GitHub repository of the standalone widget (https://github.com/jonatanalvelid/etSTED-widget/blob/main/analysis_pipelines/vesicle_proximity.py). Pre-processing, peak detection and track connection work similarly as in dynamin_rise, but the tracks are thereafter handled differently. The event detection is performed as a check of a set of conditions on pairs of tracks. Events are detected when five conditions are met: one track disappears; another track is close by; both tracks are consistently present; and at least one track has moved an accumulated vectorial distance and an absolute distance above certain thresholds. Each condition has a set of user-definable thresholds and ratios. Threshold values will expectedly vary with the type of vesicle investigated, given that various vesicles differ in morphology, motility, dynamics and density. Also, a substantial amount of this pipeline can run on the GPU, and the pipeline runs in 40–110 ms for an $800 \times 800$ pixels widefield image, depending on the number of tracked vesicles.

**Active synapses synaptotagmin-1 cluster analysis.** Analysis of synaptic vesicle clusters in each etSTED timelapse and manual STED timelapse of active synapses was performed using the scripts provided in the Code Availability section. To extract the clusters in each frame fairly, a histogram-matching bleach correction step was performed on the timelapses. Each frame was smoothed with 1 pixel Gaussian smoothing, binarized with a timelapse-constant global intensity threshold, and 1 pixel eroded once. Binary objects larger than $0.015\,\mu m^2$ were analyzed for area, aspect ratio and centroid. In the resulting data, cluster traces throughout the timelapses were connected using the centroids and a minimum Euclidean distance approach with an upper limit of $0.3\,\mu m$ movement per frame. Centroid traces of each cluster were extracted, and for the largest cluster in the first frame in each timelapse the trace was further analyzed to extract the mean square displacement (MSD). The MSD was calculated for a specific $\Delta t$ as the mean Euclidean distance between each centroid position at $t$ and $t + \Delta t$.

**Primary neuronal culture.** Primary neuronal cultures were prepared from embryonic day 18 Sprague–Dawley rat embryos. Pregnant mothers were killed with $CO_2$ inhalation and aorta cut, and brains were extracted from the embryos. Hippocampi were dissected and mechanically dissociated in MEM (Thermo Fisher Scientific, 21090022). A total of $2 \times 10^5$ cells per 60 mm culture dish were seeded on poly-D-ornithine (Sigma Aldrich, P8638) coated no. 1.5 18 mm glass coverslips (Marienfeld, 0117580), and were left to attach in MEM with 10% horse serum (Thermo Fisher Scientific, 26050088), 2 mM L-Glut (Thermo Fisher Scientific, 25030024) and 1 mM sodium pyruvate (Thermo Fisher Scientific, 11360070) at 37 °C, 95–98% humidity and 5% $CO_2$. After 2–4 h the coverslips were flipped over an astroglial feeder layer (grown in MEM supplemented with 10% horse serum, 0.6% glucose and 1% penicillin–streptomycin) and maintained in Neurobasal (Thermo Fisher Scientific, 21103049) supplemented with 2% B-27 (Thermo Fisher Scientific, 17504044), 2 mM L-glutamine and 1% penicillin–streptomycin. The cultures were treated with 5 μM 5-fluorodeoxyuridine at 2–3 days in vitro (DIV) to prevent glia overgrowth. The cultures were kept for up to 24 days and fed twice per week by replacing one-third of the medium per well: before DIV7 with Neurobasal complete medium, and from DIV7 with Braiphys (STEMCELL Technologies, 05790), 1% Pen/Strep (Gibco, 15140–114) and SM1 Supplement (STEMCELL Technologies, 05711). Experiments were performed on mature cultures at DIV14–21. All experiments were performed in accordance with animal welfare guidelines set forth by Karolinska Institute and were approved by Stockholm North Ethical Evaluation Board for Animal Research. Rats were housed with food and water available ad libitum in a 12 h light–dark environment.

**HeLa culture.** HeLa (ATCC CCL-2) cells were cultured in DMEM (Thermo Fisher Scientific, 41966029) supplemented with 10% (vol/vol) fetal bovine serum (Thermo Fisher Scientific, 10270106), 1% penicillin–streptomycin (Sigma Aldrich, P4333) and maintained at 37 °C and 5% $CO_2$ in a humidified incubator. Cells were plated on no. 1.5 18 mm glass coverslips (Marienfeld, 0117580) 24–48 h before imaging.

**HeLa transfections.** For transfection, $2 \times 10^5$ cells per well were seeded on coverslips in a 12-well plate. After 1 day the cells were transfected using FuGENE (Promega, E2311) according to the manufacturer's instructions. At 24 h after transfection the cells were washed in PBS solution, placed with phenol red-free Leibovitz's L-15 Medium (Thermo Fisher Scientific, 21083027) in a chamber and imaged.

**Neuron transfections.** Primary neurons (DIV8–14) were transfected using calcium phosphate co-precipitation protocol as reported[49]. In brief, DNA (2 μg) was diluted in TE solution (Tris-HCl, pH 7.5, 10 mM; EDTA, pH 8.0, 1 mM). $CaCl_2$ (2.5 M in 10 mM HEPES) was added to a final concentration of 250 mM. The mixed solution was added to 2× HEBS (HEPES buffered saline, pH 7.2). Neurons were pre-incubated in 200 μl conditioned medium from their culture dish with 50 μl 5× kynurenic acid stock (10 mM dissolved in unsupplemented culture medium) in a well of sterile MW12 and placed back in the incubator until the precipitate was ready. The precipitate was then added dropwise to the cells and incubated for 3–4 h. To stop the transfection, a 5:1 mix of Neurobasal medium without glutamate and kynurenic acid was pre-warmed. Then, 5 M HCl was added until the solution turned yellow. After removal of the transfection medium, the acidic medium was added to each coverslip, which was further incubated at 37 °C and 5% $CO_2$ for 15–20 min. After the incubation period the neurons were transferred back to the original Petri dish containing the conditioned medium and the construct was left to express for 18–24 h at 37 °C and 5% $CO_2$.

**Sample labeling.** For the labeling of active synapses, 1 μl Synaptotagmin-1 antibody luminal domain (1 mg ml$^{-1}$, Synaptic Systems, 105 3FB) and 1 μl FluoTag-X2 anti-mouse Ig kappa light chain nanobody conjugated to Abberior STAR635P (5 μM, NanoTag Biotechnologies, N1202-Ab635P) were pre-incubated with 98 μl pre-conditioned neuronal medium and incubated at 23 °C for 20 min. Neurons were then incubated with the Synaptotagmin-1 labeling solution for 30 min in a humidified chamber at 37 °C. After the incubation time the neurons

were left to recover for 5 min in their original medium and washed twice with artificial cerebrospinal fluid (ACSF) before imaging. Imaging was performed in ACSF at room temperature.

The labeling of F-actin and tubulin was performed as previously described[50], using live-cell fluorogenic labeling probe kits. A total of 1 mM stock solution was obtained by dissolving 50 nmol SiR-actin (Spirochrome, SC001) or SiR-tubulin (Spirochrome, SC002) in 50 μl anhydrous dimethylsulfoxide (DMSO). For labeling, the stock solution was diluted 1:1,000 for a final 1 μM staining solution, in ACSF for neurons and in cell medium for HeLa cells. Neuronal cultures were incubated for 30 min at 37 °C with the dilution, and washed twice in ACSF prior to imaging. HeLa cells were incubated for 30–45 min at 37 °C with the dilution, and washed once in cell medium prior to imaging.

For calcium imaging, a fluorescent calcium chelator labeling was used. A total of 10 μl pluronic acid F-127 solution (0.2 g pluronic acid in 1 ml DMSO, shaken at 40 °C for 20 min) was added to 50 μg Oregon Green 488 BAPTA-1, AM ester (Thermo Fisher Scientific, O6807) for a 1 mM stock solution. The neuronal culture or HeLa culture was incubated for 30 min at 37 °C with 1 μM Oregon Green 488 BAPTA-1, AM ester (1:1,000 dilution in cell medium), and washed once in ACSF or cell medium prior to imaging.

To label cholesterol, HeLa cells and primary neuronal cultures were incubated for 1 h at 37 °C and 5% $CO_2$ with Abberior STAR RED Cholesterol-PEG(1000) (1 mg ml$^{-1}$ in DMSO to a final concentration of 1 μl ml$^{-1}$). To label sphingolipids, primary neuronal cultures were incubated for 1 h at 37 °C and 5% $CO_2$ with Abberior STAR RED C12 Sphingosyl PE (d17:1/12:20) (5 mg ml$^{-1}$ in DMSO to a final concentration of 5 μl ml$^{-1}$). HeLa cells were then washed in Leibovitz's L-15 PBS and the neurons in ACSF prior to imaging.

**Plasmids.** pEGFP-N1 Dynamin1 wild type was a gift from J. Taraska (research resource identifier (RRID): Addgene_120313)[51]. pCMV-Sport6-CD63-pHluorin was a gift from D. M. Pegtel (RRID: Addgene_130901)[38]. CD63_OHu03119C_pcDNA3.1(+)-C-eGFP was obtained from GenScript Biotech.

**Imaging conditions, acquisition parameters and data visualization.** Widefield images were recorded using a 20–100 ms exposure time and a frame rate of 3.3–20 Hz. The 488 nm laser power used was 0.3–0.9 mW for neurons and 0.6–1.9 mW for HeLa cells. STED images were recorded using a pixel size of 25–30 nm, a dwell time of 30–50 μs, a 640 nm laser power of 5–16 μW and a 775 nm laser power of 59–124 mW. Exact image acquisition parameters for each experiment are listed in Supplementary Table 2. All laser powers were measured at the conjugate back focal plane of the objective lens, between the scan and the tube lens.

For visualization purposes, raw STED images have been smoothed with 0.5 pixel Gaussian smoothing. Frames from STED timelapses have been bleach corrected using histogram matching or direct-ratio methods. 3D STED images of endocytosis events have been deconvolved, as marked by the asterisks in the images, by applying a 50–60 nm (lateral) × 100 nm (axial) Lorentzian PSF and 10 iterations. Raw STED images of exocytosis have had a rolling ball background subtraction with a radius of 50 pixels applied. Frames from 23 Hz STED timelapses of synaptic vesicle dynamics have been deconvolved by applying a 70 nm Gaussian PSF and 5 iterations.

Deconvolution was performed using Richardson–Lucy deconvolution, with a regularization parameter of $1 \times 10^{-10}$, in Inspector (Max-Planck Innovation). For data visualization and post-acquisition analysis, custom-written scripts and JupyterLab notebooks in Fiji (ImageJ) and Python have been used (see Code Availability), equipped with additional packages such as scikit-image[52], jupyterlab[53] and matplotlib[54].

**True event detection and detected real events quantification.** The ratios of true-positive event detections (True) and detected real events (Det), compared with the number of events, have been quantified as measures of the accuracy and precision of the rapid_synaptic_spikes analysis pipeline. The quantification was performed through full widefield timelapse recordings, without running the etSTED method, with the same acquisition parameters as in a full etSTED experiment. The timelapses were manually annotated for calcium events and also analyzed with the analysis pipeline. The results of the two were compared to calculate the two ratios. This quantification was performed in multiple experiments and multiple cells given that it depends on the fluorescence background, the cellular structure, the labeling density, the acquisition parameters and the user-inputted pipeline parameters. Plotted is one data point for each cell in the various experiments analyzed.

**Statistics.** All statistical tests are two-sample two-sided Kolmogorov–Smirnov tests, where the asterisk symbol (*) indicates $P < 0.05$ and NS indicates $P > 0.05$. Shaded confidence interval areas are chosen as the 83% level, meaning that non-overlapping areas infer a significant difference at that part of the curve.

**Reporting summary.** Further information on research design is available in the Nature Research Reporting Summary linked to this article.

## Data availability

The data that support the method implementation and support the findings in this study, including images, log files and metadata, are available at Zenodo, reference number 5593270 (ref. [55]).

## Code availability

The code of the developed widget in this study is available in the source code of the microscope control software ImSwitch at https://github.com/kasasxav/ImSwitch, specifically the widget and controller are available at https://github.com/kasasxav/ImSwitch/blob/master/imswitch/imcontrol/controller/controllers/EtSTEDController.py and https://github.com/kasasxav/ImSwitch/blob/master/imswitch/imcontrol/view/widgets/EtSTEDWidget.py. The minimal standalone widget developed to support implementation of the event-triggered method in other microscope control software is available in two versions: one with simulated experiments at https://github.com/jonatanalvelid/etSTED-widget, and one that is software-agnostic at https://github.com/jonatanalvelid/etSTED-widget-base. The code of the analysis supporting the findings in this work, as ImageJ macros and Jupyter notebooks together with example data, is available at https://github.com/jonatanalvelid/etSTEDanalysis.

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

## Acknowledgements

I.T. thanks the Swedish Foundation for Strategic Research (project FFL15–0031) and the European Union (Horizon 2020 programme, project IMAGEOMICS 964016) for supporting the research. The authors thank X. Casas Moreno for help with the control software ImSwitch and help with the implementation of the widget, and F. Pennacchietti for help with the synaptotagmin-1 image analysis.

## Author contributions

I.T. designed and supervised the research. I.T. and J.A. conceived the project idea. J.A. planned and built the set-up; developed the control software widget, the controller and the real-time analysis pipelines; performed the experiments; and performed data analysis. M.D. provided biological guidance; performed the application experiments; prepared the samples; and performed data analysis. C.S. helped with data analysis and data interpretation. J.A. and I.T. wrote the manuscript with input from all of the authors.

## Funding

## Competing interests

The authors declare no competing interests.

## Additional information

**Extended data** are available for this paper at https://doi.org/10.1038/s41592-022-01588-y.

**Correspondence and requests for materials** should be addressed to Ilaria Testa.

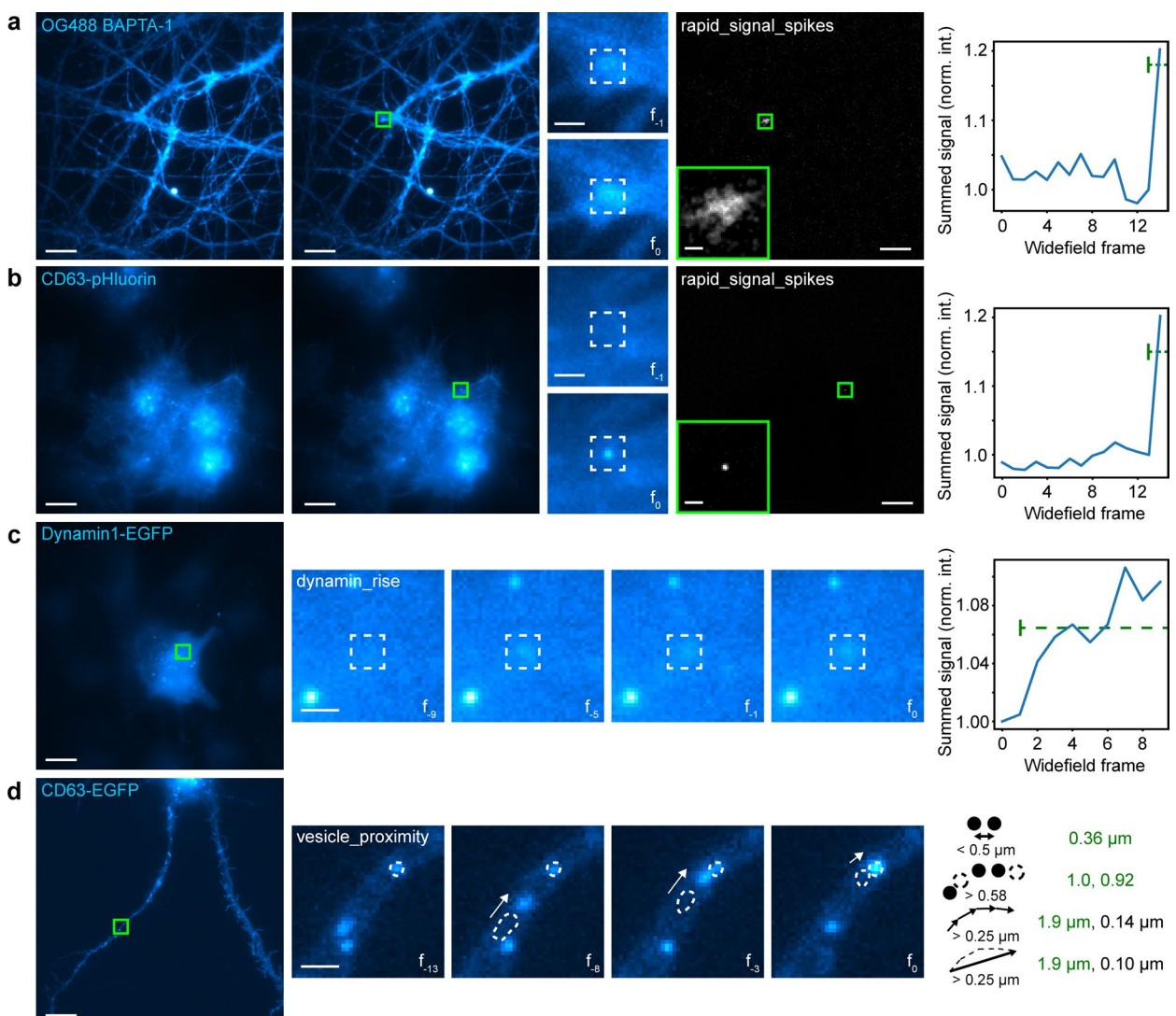

**Extended Data Fig. 1 | Applications of etSTED analysis pipelines. a**, rapid_signal_spikes applied to hippocampal neurons labeled with BAPTA-OregonGreen488 to detect calcium spikes. Representative example from N = 946 events, N = 62 cells. **b**, rapid_signal_spikes applied to HeLa cells labeled with CD63-pHluorin in order to detect fusion events of endosomes to the plasma membrane. Representative example from N = 232 events, N = 29 cells. **c**, dynamin_rise applied to HeLa cells labeled with dynamin1-GFP in order to detect endocytosis at the plasma membrane. Representative example from N = 53 events, N = 22 cells. **d**, vesicle_proximity applied to hippocampal neurons labeled with CD63-GFP in order to detect interaction of intracellular vesicles. Representative example from N = 172 events, N = 30 cells. Scale bars 10 μm (widefield and analysis), 1 μm (zooms).

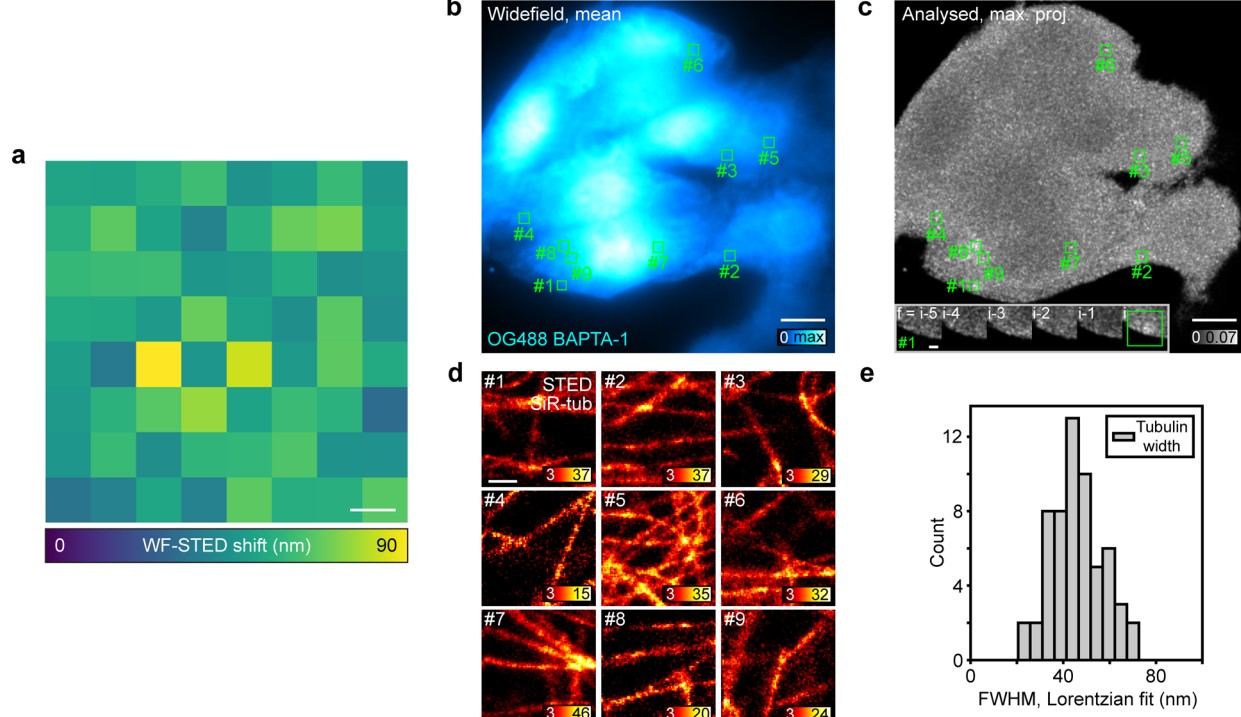

**Extended Data Fig. 2 | Characterization of etSTED: coordinate transform and etSTED experiments with calcium event triggering and STED imaging of microtubules. a**, Characterization of residual shift after coordinate transformation between widefield and scanning space across the field of view (FOV). N = 1048 beads. **b**, etSTED experiment in HeLa cells with BAPTA-1 calcium imaging with mean widefield. **c**, Maximum projection ratiometric image, showing the location of 9 detected events throughout the experiment. **d**, Event-triggered $2 \times 2 \mu m^2$ STED images of SiR-tubulin. Inset (center) shows a zoomed-in view of the ratiometric image around event #1 for the five frames leading up to the event. **e**, Characterization of the imaged microtubule size in etSTED images across multiple experiments and across the FOV, as fitted FWHM in line profiles across single microtubules. N = 69 events, N = 8 cells. Scale bars, 10 μm (**a**,**b**,**c**), 2 μm (**c** inset) and 500 nm (**d**).

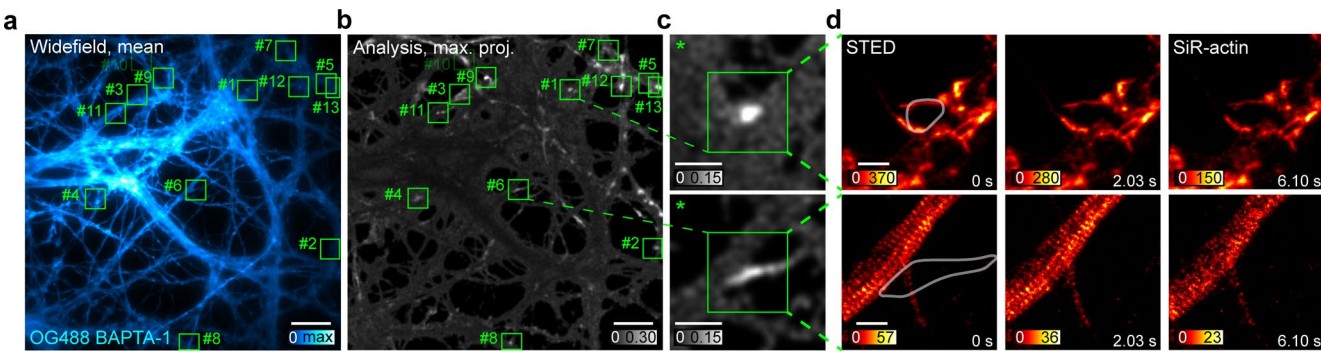

**Extended Data Fig. 3 | etSTED experiments in neurons with widefield imaging of calcium signaling (BAPTA-1) and STED timelapse imaging of actin (SiR-actin). a**, Mean widefield image of all triggering widefield frames. **b**, Maximum-projected analysis ratiometric image of all detected events in experiment. Green squares show location of detected events (**a,b**). **c**, Zoomed-in views of the ratiometric image in two detected events. **d**, 0.99 Hz etSTED timelapse of the actin structures. N = 234 events, N = 19 cells. Overlaid semi-transparent white outlines show extent of detected local calcium activity. Scale bars, 10 μm (**a,b**), 3 μm (**c**), 1 μm (**d**).

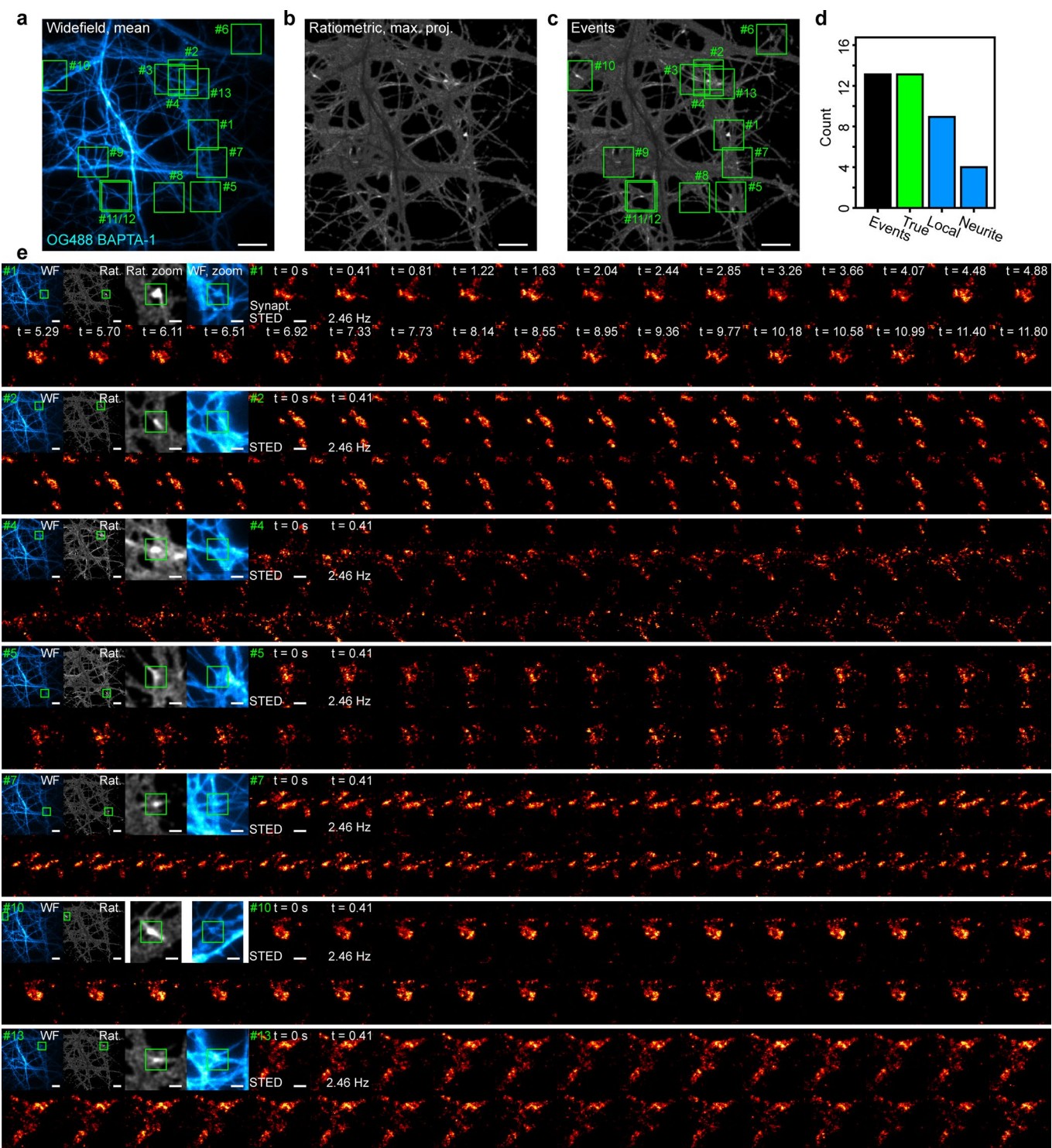

**Extended Data Fig. 4 | etSTED experiment in neurons with calcium imaging (Oregon Green 488 BAPTA-1) and STED timelapse imaging of synaptic vesicles (synaptotagmin-1_STAR635P). a**, Mean image of all widefield frames with detected events. Boxes are centered on the coordinates of true (green) and false (magenta) detected events. **b**, Maximum projection of all ratiometric preprocessed widefield frames with detected events. **c**, Same as **b** with boxes centered on the coordinates of true (green) and false (magenta) detected events. **d**, Number of detected events, true events, local events, and neurite-wide events. **e**, Widefield frame, ratiometric preprocessed frame, zoom-ins of the widefield and ratiometric, and event-triggered STED timelapse (30 frames, 0.99 Hz) of events as numbered and marked in **a**,**c**. N = 186 events, N = 17 cells. Boxes marks the center of the detected event. Same scales and time labels apply to all timelapses. Scale bars, 10 μm (**a**,**b**,**c**,**e** widefield and ratiometric), 2 μm (**e** zooms) and 1 μm (**e** STED).

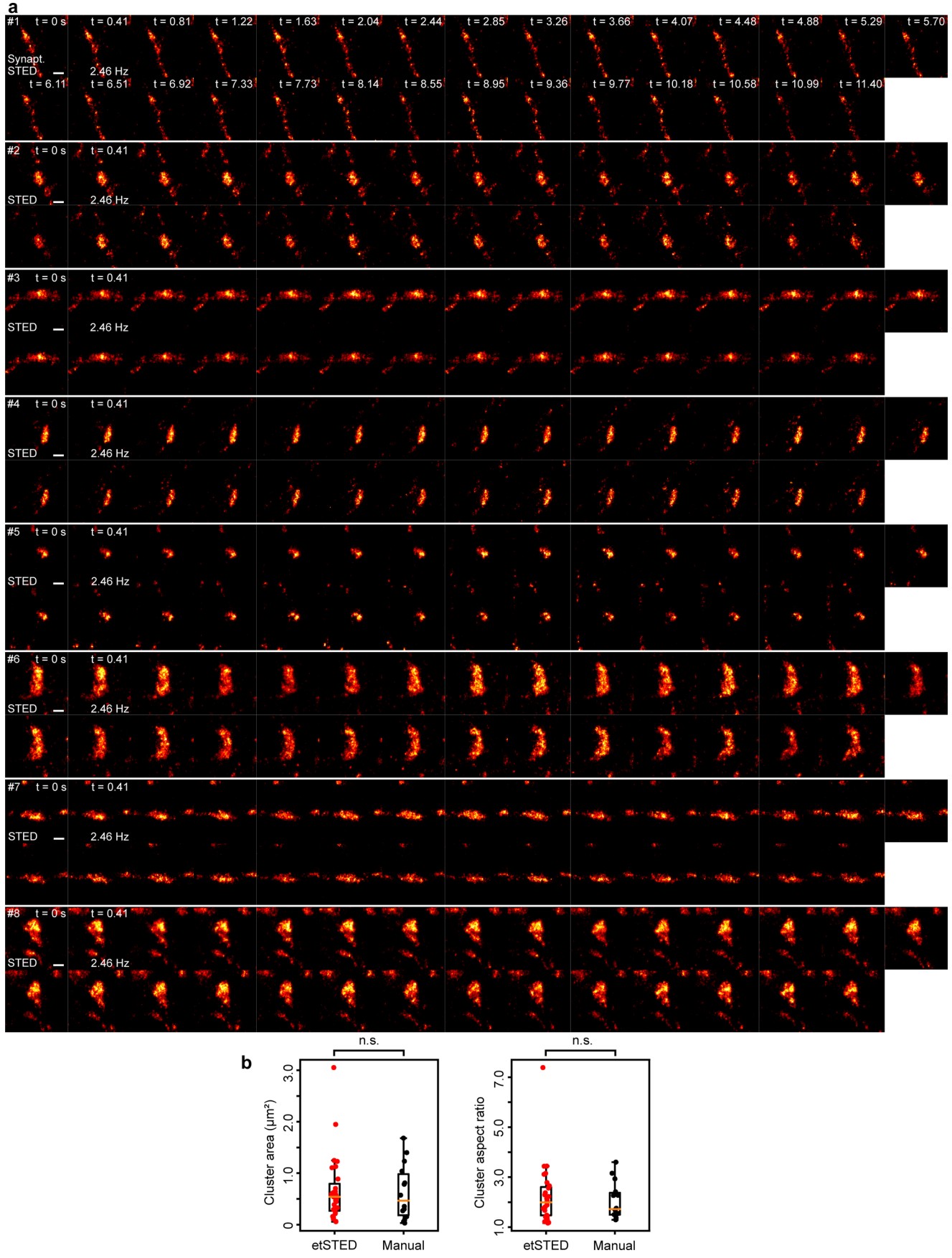

**Extended Data Fig. 5 | See next page for caption.**

**Extended Data Fig. 5 | Manual STED timelapse imaging of synaptic vesicles (synaptotagmin-1_STAR635P). a**, STED timelapses of synaptic vesicle clusters in active synapses. **b**, Distributions of area (left) and aspect ratio (right) for the clusters in calcium-activity-triggered STED timelapses (red) and manual timelapses (black). Box plots show 25–75% IQRs, middle line is mean, and whiskers reach 1.5 times beyond the first and third quartiles. N = 26 clusters in N = 10 cells (red), N = 14 clusters in N = 14 cells (black). Statistical test used is a two-sample two-sided Kolmogorov–Smirnov test: p = 0.90, test statistic = 0.17 (area); p = 0.76, test statistic = 0.20 (aspect ratio). Scale bars, 1 μm.

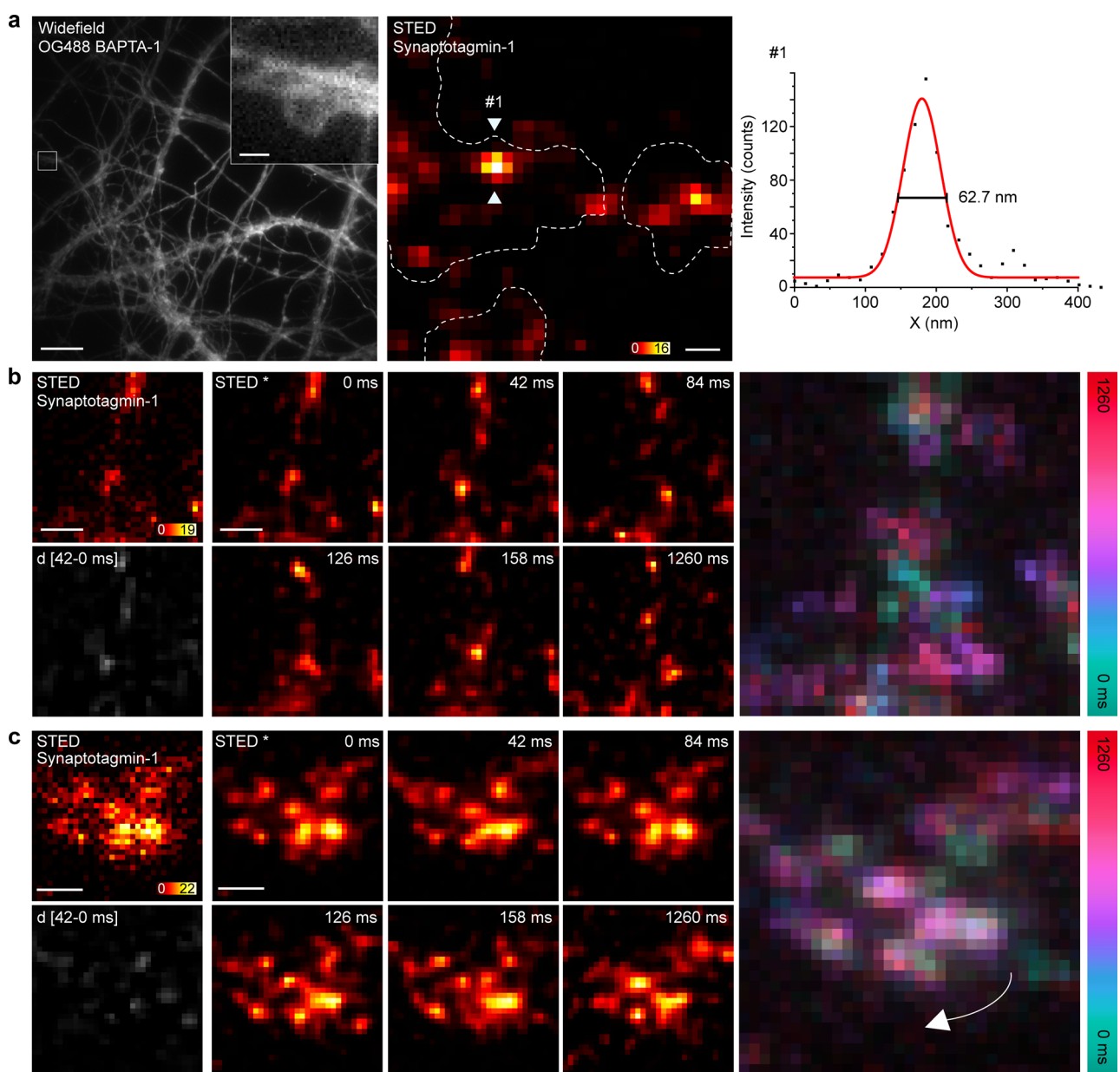

**Extended Data Fig. 6 | Imaging of synaptic vesicle dynamics with etSTED at 23 Hz. a**, (left) Oregon Green 488 BAPTA-1 widefield image. ROI I shows the neurite region in which a calcium event was detected. (right) Single frames of a STED timelapse recording showing synaptotagmin-1_STAR635P-labeled single synaptic vesicles. A line profile drawn across a synaptic vesicle shows a 62.7 nm FWHM. **b**, Representative 23.3 Hz STED timelapse recording of 30 frames showing the dynamics of single synaptic vesicles in active synapse upon a detected calcium event. In gray, the difference image between the first and second frame. The temporal-color-coded image depicts the mobility of synaptic vesicles in 1260 ms, with the white arrow displaying the directional movement. **c**, Representative 23.3 Hz STED timelapse recording of 30 frames showing the dynamics of a synaptic vesicle recycling pool in an active synapse upon a detected calcium event. In gray, the difference image between the first and second frame. The temporal-color-coded image depicts the mobility of the synaptic vesicle recycling pool in 1260 ms. N = 379 events, N = 14 cells. Scale bars, 10 μm (**a**, widefield), 1 μm (**a** widefield inset), 100 nm (**a** STED) and 250 nm (**b**,**c**).

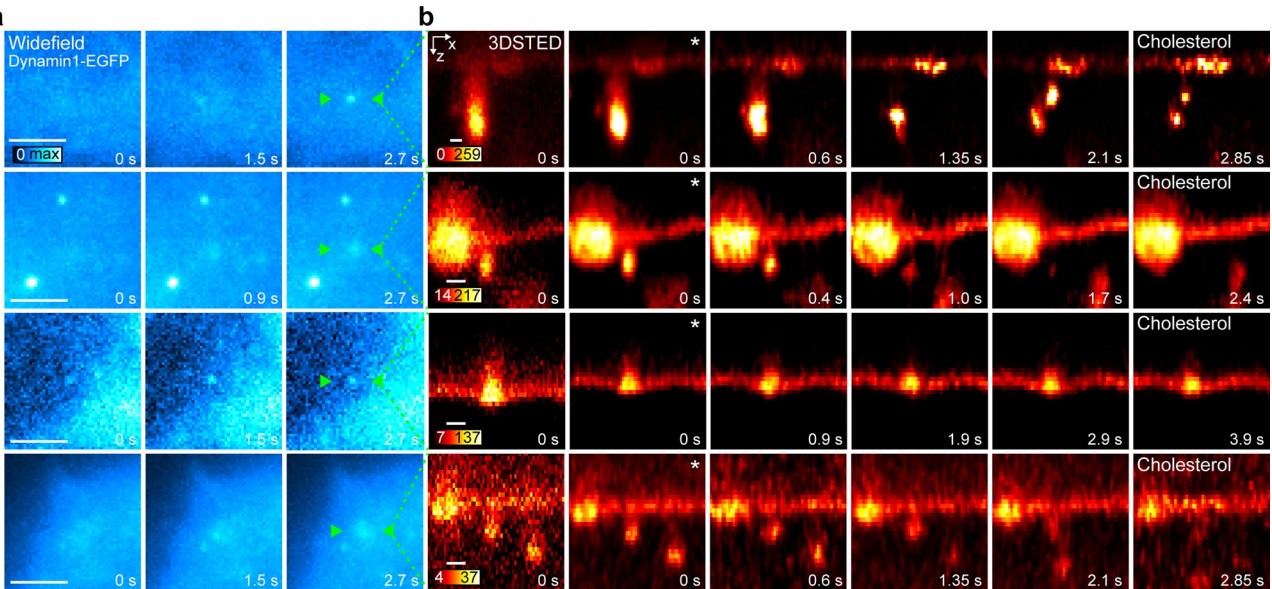

**Extended Data Fig. 7 | etSTED experiments in HeLa cells with widefield imaging of dynamin (dynamin1-EGFP) and 3D STED timelapse imaging of membrane dynamics (cholesterol-KK114) during endocytosis. a**, Widefield images leading up to the triggering frame (last image), showing the slow rise in fluorescence signal signifying recruitment of dynamin at the site. **b**, Representative frames from 5.9 Hz 3D STED timelapses of XZ cross-sections across the detected event, showing various membrane dynamics during the endocytosis events taking place. Arrows marks the scanned section in X (**a**). N = 53 events, N = 22 cells. Same scales apply to all frames of the timelapses. Asterisks (*) indicate deconvolved frames, and applies to all following frames in the same timelapse. Scale bars, 2 μm (**a**), 250 nm (**b**).

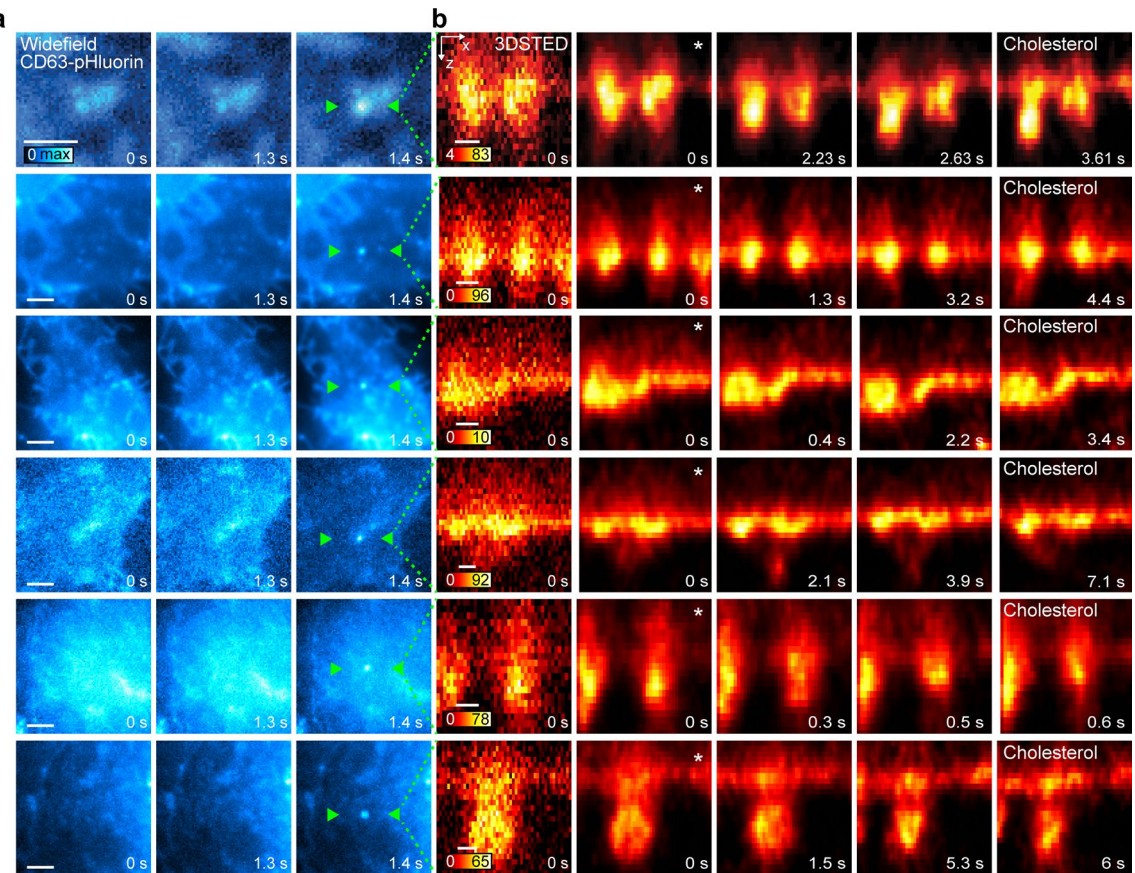

**Extended Data Fig. 8 | etSTED experiments in HeLa cells with widefield imaging of CD63-pHluorin and 3D STED timelapse imaging of membrane dynamics (cholesterol-KK114) during exocytosis. a**, Widefield images leading up to the triggering frame (last image), showing the rapidly appearing (frame-to-frame) fluorescence signal signifying a local pH neutralization at the site. **b**, Representative frames from 11 Hz 3D STED timelapses of XZ cross-sections across the detected event, showing various membrane dynamics during the exocytosis events taking place. Arrows marks the scanned section in X (**a**). N = 232 events, N = 29 cells. Same scales apply to all frames of the timelapses. Asterisks (*) indicate deconvolved frames, and applies to all following frames in the same timelapse. Scale bars, 2 μm (**a**), 250 nm (**b**).

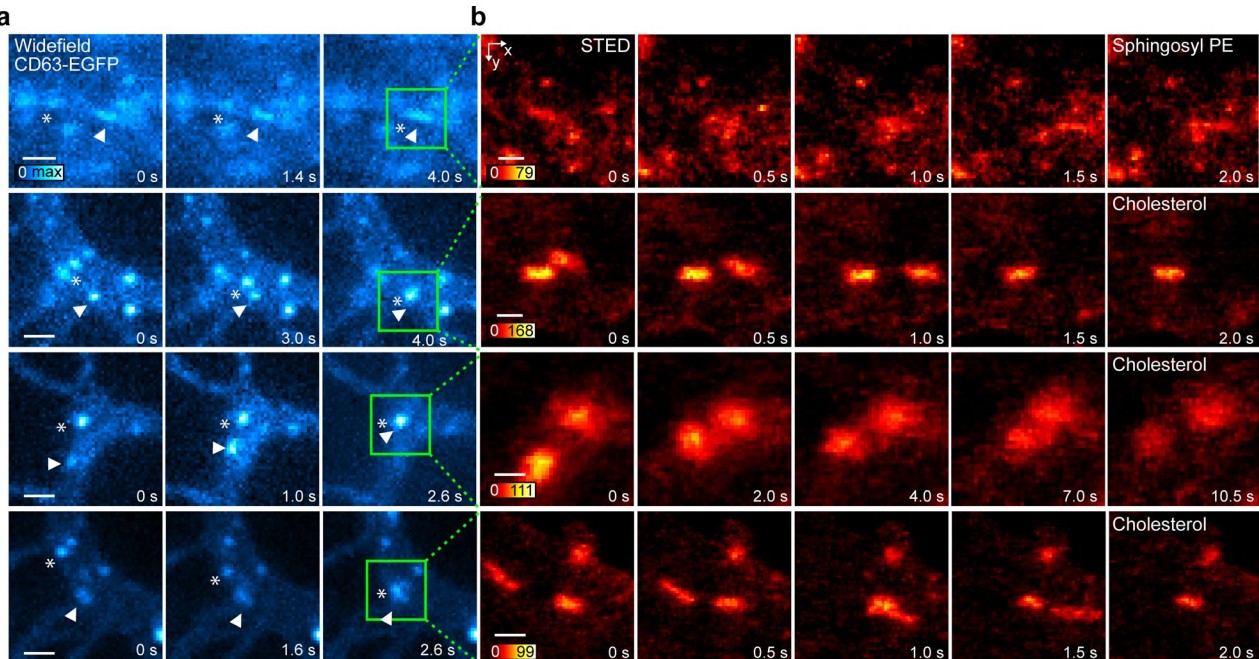

**Extended Data Fig. 9 | etSTED experiments in HeLa cells with widefield imaging of intracellular vesicles (CD63-EGFP) and STED timelapse imaging of membrane dynamics (cholesterol-KK114) during intracellular vesicle interaction. a**, Widefield images leading up to the triggering frame (last image), showing the increasing proximity of two vesicles (asterisk (mobile) and arrow (stationary)) indicating a potential interaction. **b**, Representative frames from 2.8 Hz STED timelapses around the detected event, showing the interaction of two vesicles, labeled with either sphingosyl-PE_Abberior STAR RED or cholesterol-Abberior STAR RED. Boxes marks the STED imaged area (**a**). Sphingosyl PE: N = 123 events, N = 22 cells; cholesterol: N = 49 events, N = 8 cells. Same scales and time labels apply to all timelapses. Asterisks (*) indicates deconvolved frames, and applies to all following frames in the same timelapse. Scale bars, 2 μm (**a**), 250 nm (**b**).

# Reporting Summary

## Statistics

For all statistical analyses, confirm that the following items are present in the figure legend, table legend, main text, or Methods section.

| n/a | Confirmed | |
|---|---|---|
| ☐ | ☒ | The exact sample size (*n*) for each experimental group/condition, given as a discrete number and unit of measurement |
| ☐ | ☒ | A statement on whether measurements were taken from distinct samples or whether the same sample was measured repeatedly |
| ☐ | ☒ | The statistical test(s) used AND whether they are one- or two-sided *Only common tests should be described solely by name; describe more complex techniques in the Methods section.* |
| ☒ | ☐ | A description of all covariates tested |
| ☒ | ☐ | A description of any assumptions or corrections, such as tests of normality and adjustment for multiple comparisons |
| ☐ | ☒ | A full description of the statistical parameters including central tendency (e.g. means) or other basic estimates (e.g. regression coefficient) AND variation (e.g. standard deviation) or associated estimates of uncertainty (e.g. confidence intervals) |
| ☐ | ☒ | For null hypothesis testing, the test statistic (e.g. *F*, *t*, *r*) with confidence intervals, effect sizes, degrees of freedom and *P* value noted *Give P values as exact values whenever suitable.* |
| ☒ | ☐ | For Bayesian analysis, information on the choice of priors and Markov chain Monte Carlo settings |
| ☒ | ☐ | For hierarchical and complex designs, identification of the appropriate level for tests and full reporting of outcomes |
| ☒ | ☐ | Estimates of effect sizes (e.g. Cohen's *d*, Pearson's *r*), indicating how they were calculated |

*Our web collection on statistics for biologists contains articles on many of the points above.*

## Software and code

Policy information about availability of computer code

| | |
|---|---|
| Data collection | The imaging was performed with open-source hardware and acquisition control software ImSwitch in Python, developed in the lab and available at https://github.com/kasasxav/ImSwitch (v1.3.0) and https://github.com/jonatanalvelid/ImSwitch-etSTED (v1.0). The etSTED acquisition method was implemented as a widget in ImSwitch, available as the ImSwitch-adapted widget at https://github.com/etSTED-widget (v1.0), as a standalone software-agnostic widget at https://github.com/jonatanalvelid/etSTED-widget-base (v1.0), and in the main ImSwitch repository. |
| Data analysis | Real-time analysis and post-acquisition data analysis was performed with custom-written scripts and JupyterLab notebooks in Fiji (ImageJ v1.53c) and Python (v3.7-v3.9). Overall, the control software and analysis was performed with the following Python packages: numpy (v1.19), scipy (v1.6), scikit-image (v0.18), cupy-cuda100 (v9.0.0), matplotlib (v3.4.1), opencv (v4.5.2.54), trackpy (v0.5.0), pandas (v1.3.2), pyqtgraph (v0.12.3), napari (v0.4.7), and jupyterlab (v3.2.9). For deconvolution, the Imspector software (Max-Planck Innovation) was used.

Post-acquisition analysis scripts are available at https://github.com/jonatanalvelid/etSTEDanalysis. Real-time image analysis during etSTED acquisition was performed using three optimized image analysis pipelines: rapid_signal_spikes (https://github.com/jonatanalvelid/etSTED-widget/blob/main/analysis_pipelines/rapid_signal_spikes.py), dynamin_rise (https://github.com/jonatanalvelid/etSTED-widget/blob/main/analysis_pipelines/dynamin_rise.py), and vesicle_proximity (https://github.com/jonatanalvelid/etSTED-widget/blob/main/analysis_pipelines/vesicle_proximity.py). |

For manuscripts utilizing custom algorithms or software that are central to the research but not yet described in published literature, software must be made available to editors and reviewers. We strongly encourage code deposition in a community repository (e.g. GitHub). See the Nature Portfolio guidelines for submitting code & software for further information.

## Data

Policy information about <u>availability of data</u>

All manuscripts must include a <u>data availability statement</u>. This statement should provide the following information, where applicable:

- Accession codes, unique identifiers, or web links for publicly available datasets
- A description of any restrictions on data availability
- For clinical datasets or third party data, please ensure that the statement adheres to our <u>policy</u>

> The data that support the implementation of the method and support the findings in this study, including images, log-files, and metadata, are openly available in Zenodo at https://doi.org/10.5281/zenodo.5593270, reference number 5593270.

# Field-specific reporting

Please select the one below that is the best fit for your research. If you are not sure, read the appropriate sections before making your selection.

☒ Life sciences  ☐ Behavioural & social sciences  ☐ Ecological, evolutionary & environmental sciences

For a reference copy of the document with all sections, see nature.com/documents/nr-reporting-summary-flat.pdf

# Life sciences study design

All studies must disclose on these points even when the disclosure is negative.

| | |
|---|---|
| Sample size | No sample size dependent statistical testing was performed. Sample sizes for the different experiments were chosen to multiple experimental days and cover glass replications (N = 2–7), many cell replications on each glass (N = 2–9, N = 8–62 cells in total), and on average many triggered or manual images/timelapses/events in each cell (N = 2–37, N = 14–379 events in total); see figure legends for specific numbers for each experiment. This ensures a large enough statistical ground during statistical testing, and was chosen in order to make sure to contain recorded images with similar image quality and resolution independently from the cell, glass or experiment. |
| Data exclusions | No data was excluded from the analysis. |
| Replication | The STED imaging quality was reproducible in different cells, different days of recording and for extended times after system alignment. System alignment was stable over time for the full day of recording. Experiments were replicated and performed independently multiple times inside a time period of half a year. All attempts at replication were successful. |
| Randomization | No allocation into experimental groups was performed. |
| Blinding | No allocation into experimental groups was performed. |

# Reporting for specific materials, systems and methods

We require information from authors about some types of materials, experimental systems and methods used in many studies. Here, indicate whether each material, system or method listed is relevant to your study. If you are not sure if a list item applies to your research, read the appropriate section before selecting a response.

### Materials & experimental systems

| n/a | Involved in the study |
|---|---|
| ☐ | ☒ Antibodies |
| ☐ | ☒ Eukaryotic cell lines |
| ☒ | ☐ Palaeontology and archaeology |
| ☐ | ☒ Animals and other organisms |
| ☒ | ☐ Human research participants |
| ☒ | ☐ Clinical data |
| ☒ | ☐ Dual use research of concern |

### Methods

| n/a | Involved in the study |
|---|---|
| ☒ | ☐ ChIP-seq |
| ☒ | ☐ Flow cytometry |
| ☒ | ☐ MRI-based neuroimaging |

## Antibodies

| | |
|---|---|
| Antibodies used | Synaptotagmin-1 antibody luminal domain (Synaptic Systems, cat. no. 105 3FB); FluoTag-X2 anti-mouse Ig kappa light chain nanobody conjugated to Abberior STAR635P (NanoTag Biotechnologies, cat. no. N1202-Ab635P). |
| Validation | The antibody against synaptotagmin-1 has been tested and used in many publications over the past 20 years. Here are a few examples: (1) Geppert M., Annual review of neuroscience (1998) 21: 75-95, (2) Jahn R., Annual review of neuroscience (1994) 17: |

219-46, (3) Südhof TC, Nature (1995) 3756533: 645-53. We think that a formal validation is not necessary in this case.

## Eukaryotic cell lines

Policy information about cell lines

| | |
|---|---|
| Cell line source(s) | HeLa, ATCC CCL-2 |
| Authentication | None of the cell lines used were authenticated. |
| Mycoplasma contamination | No testing for mycoplasma contamination. |
| Commonly misidentified lines (See ICLAC register) | No commonly misidentified cell lines were used in this study. |

## Animals and other organisms

Policy information about studies involving animals; ARRIVE guidelines recommended for reporting animal research

| | |
|---|---|
| Laboratory animals | Sprague Dawley rat, embryonic day 18 |
| Wild animals | The study did not involve wild animals. |
| Field-collected samples | The study did not involve field-collected samples. |
| Ethics oversight | All experiments were performed in accordance with animal welfare guidelines set forth by Karolinska Institutet and were approved by Stockholm North Ethical Evaluation Board for Animal Research. |

Note that full information on the approval of the study protocol must also be provided in the manuscript.

