## [Peer Review File · Nature Methods]

Peer Review Information

Manuscript Title: Event-triggered STED imaging

Corresponding author name(s): Ilaria Testa

Editorial Notes:

Reviewer Comments & Decisions:

Decision Letter, initial version:

Dear Ilaria,

Season's greetings!

Your Brief Communication, "Event-triggered STED imaging", has now been seen by three reviewers. As you will see from their comments below, although the reviewers find your work of considerable potential interest, they have raised a number of concerns. We are interested in the possibility of publishing your paper in Nature Methods, but would like to consider your response to these concerns before we reach a final decision on publication.

As you will note, the referees have some technical concerns regarding your claimed achievements and the limits of your approach in terms of temporal resolution. They also raise concerns about general applicability. To address this, we would like to see an additional demonstration on a distinct sample if possible.

In terms of changes to the paper itself, they would like the work better placed in context. In addition, they are concerned that the method represents more of an achievement than a new method, and referee 3 is concerned that it would be easier for an interested user to start from scratch than to try to implement your approach. In response to this concern, we ask that you provide some sort of template for users to implement the general strategy you've used that is as microscope and software agnostic as is reasonable.

We are committed to providing a fair and constructive peer-review process. Do not hesitate to contact us if there are specific requests from the reviewers that you believe are technically impossible or unlikely to yield a meaningful outcome. Please also let us know if you will be unable to add a second demonstration imaging a distinct biological phenomenon in a reasonable time.

REDACTED

We hope to receive your revised paper within XX weeks [**ED TO CUSTOMIZE AS NEEDED**]. If you cannot send it within this time, please let us know. In this event, we will still be happy to reconsider

your paper at a later date so long as nothing similar has been accepted for publication at Nature Methods or published elsewhere.

OPEN SCIENCE REQUIREMENTS

REPORTING SUMMARY AND EDITORIAL POLICY CHECKLISTS

DATA AVAILABILITY

We strongly encourage you to deposit all new data associated with the paper in a persistent repository where they can be freely and enduringly accessed. We recommend submitting the data to discipline-specific and community-recognized repositories; a list of repositories is provided here:

<http://www.nature.com/sdata/policies/repositories>

All novel DNA and RNA sequencing data, protein sequences, genetic polymorphisms, linked genotype and phenotype data, gene expression data, macromolecular structures, and proteomics data must be deposited in a publicly accessible database, and accession codes and associated hyperlinks must be provided in the "Data Availability" section.

Please include a "Data availability" subsection in the Online Methods. This section should inform readers about the availability of the data used to support the conclusions of your study, including accession codes to public repositories, references to source data that may be published alongside the

paper, unique identifiers such as URLs to data repository entries, or data set DOIs, and any other statement about data availability. At a minimum, you should include the following statement: "The data that support the findings of this study are available from the corresponding author upon request", describing which data is available upon request and mentioning any restrictions on availability. If DOIs are provided, please include these in the Reference list (authors, title, publisher (repository name), identifier, year). For more guidance on how to write this section please see: <http://www.nature.com/authors/policies/data/data-availability-statements-data-citations.pdf>

CODE AVAILABILITY

Please include a "Code Availability" subsection in the Online Methods which details how your custom code is made available. Only in rare cases (where code is not central to the main conclusions of the paper) is the statement "available upon request" allowed (and reasons should be specified).

For more information on our code sharing policy and requirements, please see: <https://www.nature.com/nature-research/editorial-policies/reporting-standards#availability-of-computer-code>

MATERIALS AVAILABILITY

ORCID

Nature Methods is committed to improving transparency in authorship. As part of our efforts in this direction, we are now requesting that all authors identified as 'corresponding author' on published papers create and link their Open Researcher and Contributor Identifier (ORCID) with their account on the Manuscript Tracking System (MTS), prior to acceptance. This applies to primary research papers only. ORCID helps the scientific community achieve unambiguous attribution of all scholarly contributions. You can create and link your ORCID from the home page of the MTS by clicking on 'Modify my Springer Nature account'. For more information please visit <http://www.springernature.com/orcid>.

Sincerely,
Rita

Rita Strack, Ph.D.

Senior Editor
Nature Methods

Reviewers' Comments:

Reviewer #1:

Remarks to the Author:

This paper describes a very cool technique that uses automated image analysis to perform multi-scale live-cell microscopy. Specifically, the authors analyze calcium dynamics in a widefield fluorescence microscope, then zoom in on areas of interest to get super resolution STED images at areas where calcium signalling is occurring. They demonstrate this by taking super-resolution images of the cytoskeleton and a synaptic protein at synapses. It is an overall well-written paper with claims that are well-supported by their data.

The authors do a good job at describing the significance of their work, which in my estimation is: 1) They can make better use of a sample's photon budget by only doing super-resolution imaging at the areas of interest. 2) By doing so they can collect more data of rare events. This seems like it would be a very useful technique, but the application areas are far from my own expertise, so I am probably not the best judge of its potential impact on the field.

The authors also claim that their method will be applicable to other combinations of fast and high-resolution methods. I think this could use a little clarification. Their specific technique would be hard to generalize as-is, because it relies on application-specific things like the analysis of calcium signals and coordinate transformations that are (as far as I know) specific to STED. The broader concept (automated image-adaptive resolution changing or instrument control) can certainly be generalized to other types of experiments, but I don't think this is the first paper to demonstrate that concept (see comments below).

I see no major flaws with their technical analysis, other than some areas described below for improving the clarity of its presentation

I hope my comments are helpful in improving the quality of the manuscript.

Sincerely,
Henry Pinkard
UC Berkeley

Specific comments

Abstract

I would remove the word "instantly" if what you really mean is less than 40ms

It might be nice to have another clause or sentence at the end saying what you used your method for and what its implications are

Page 1

"and tissues even dynamically" -- reads confusingly. Missing comma?

"its most common point-scanning" -- does this mean it can be implemented in non point scanning mode? please clarify

It might be worth discussing adaptive approaches as a whole, not just in STED. For example:

Scherf, N. & Huisken, J. The smart and gentle microscope. Nat. Biotechnol. 33, 815–818 (2015).

Li, B., Wu, C., Wang, M., Charan, K. & Xu, C. An adaptive excitation source for high-speed multiphoton microscopy. *Nat. Methods* 17, 163–166 (2020).
 Chu, K. K., Lim, D. & Mertz, J. Two-photon microscopy with adaptive illumination power. *Biomedical Optics BIOMED* 2008, 1–3 (2008).
 Pinkard, H., Baghdassarian, H., Mujal, A. et al. Learned adaptive multiphoton illumination microscopy for large-scale immune response imaging. *Nat Commun* 12, 1916 (2021). [Disclosure: my work]
 Yan X, Stuurman N, Ribeiro SA, Tanenbaum ME, Horlbeck MA, Liem CR, Jost M, Weissman JS, Vale RD. High-content imaging-based pooled CRISPR screens in mammalian cells. *J Cell Biol.* 2021 Feb 1;220(2):e202008158. doi: 10.1083/jcb.202008158. PMID: 33465779; PMCID: PMC7821101.

Page 2

facilitate  facilitates

flexibly optimized to the problem at hand  what does this mean? can you clarify?

events such as localized calcium spikes in neurons  is this the particular event you're detecting or just one of many?

Thanks to the generalized implementation of etSTED  comma
 other combination of dynamic events and structures can be observed -- you mean they can be used a trigger?

Furthermore, the adaptive method does not lend itself limited to use specifically  confusingly written

but instead the concept is applicable to other combinations of fast and high-resolution imaging methods.  yes, and many already have, as described above

"an optimized analysis pipeline"  what does this mean? What would a non-optimized analysis pipeline be like??

" to confirm the validity of the event in a post-acquisition analysis "  does this mean by a human?

"The method can run indefinitely, and the focus lock in place in the microscope5 (Supplementary Fig. 2) maintains a stable sample throughout the experiment and ensures that the widefield and STED images are recorded in the same focal plane."  why would you need the focus lock to ensure this? Are you switching objectives?

page 3

" By applying the analysis pipeline post-acquisition on a widefield timelapse recording (Fig. 2a) we could extract calcium intensity traces at the sites of detected events also after the detection (Fig. 2b, left). We can confirm that the detected events are, on average, in 91% (0.91 ± 0.19) of the cases true calcium spike events (Fig. 2c), as the characteristic initial spike and decay across hundreds of milliseconds can be seen"  This sounds like you're applying the same analysis algorithm offline and online. I'm guessing I'm missing something because one wouldnt expect different results doing that. Can you clarify that?

" Important to note is that these events can be sorted out in post-acquisition analysis by inspecting the calcium trace up to the point of event detection " Why wouldnt you just use this info in real time if you have it?

page 5

"inside < 40 ms"  "with 40 ms"

"proving the potential for correlating millisecond dynamics such as calcium sensing in living cells with super-resolved details of proteins of interest."  maybe make this more clear that's just referring to the data collection itself, not analysis you did on it

" in the order of 100–1000x faster" -- citation or calculation supporting this?

"Moreover, it allows experiments not previously feasible, where the dynamics and structure of synaptic vesicle pools can be directly correlated to the presence or lack of local calcium oscillations" -- might be a good place to talk about biases your method could introduce

"hile the method as proposed here combines widefield imaging of a calcium sensor with STED imaging of actin, microtubules, or synaptic vesicles, the possibilities to generalize and extend the method are endless." -- I think this is not the first and only implementation of this general idea, and so others should be cited

"to the accuracy and speed of liking." Not exactly true. Some of the analysis routines you described just above this might take a nontrivial amount of computation time

Figure 1

You shouldnt use red and green to differentiate things, as they appear identical for the most common type of color blindness. For a great discussion of this see:
<https://www.nature.com/articles/s41467-020-19160-7>

b) "Calc" is not described in the caption

c) Those inset images are incredibly tiny, and I'm not sure what im supposed to take away from them

d) Lots of acronyms on this that aren't defined anywhere. Also, theres a lot of detail that I think is not releveant to the main point of the paper. Maybe it makes sense to have this as a supp figure, where it can be fully explained, and have a simplified block diagram in this figure

e/f) These seem to be technical details that could perhaps be moved to the supplement. It seems to me that the coordinate shift is more of an implementation detail than a core part of the method. I don't understand why you need to prove why the STED resolution would be affected by the triggering method. Also, there's no control to compare to here. Also, this is really the width of a microtubule imaged under STED, no the actual resolution, right?

Figure 2

There is a ton of information in here, and a lot of text packed in. I think it could benefit a lot from trimming down to what you think are the most important takeaways for the reader, and moving other stuff to supplement. It also seems like there are several places where you have abbreviations that you dont define in the caption, which makes the figure a lot harder to understand

b) Define dF/F . State what annotated event coordinates are in the caption (in addition to text)

c) I'm confused as to why this is a distribution, rather than just a number with some error bars. My understanding is that this is meant to quantify the accuracy of your detection algorithm. Would something like an ROC curve or confusion matrix be a better description of this?

d) Again, don't use red/green

f) Overlapping text on leftmost picture makes it hard to read. I dont think you need "Rat zoom". You could make arrows going to where it came from in larger FOV. You dont describe what the white outlines are in the caption

Reviewer #2:

Remarks to the Author:

In this manuscript, Alvelid and coworkers describe a framework that allows multiscale optical interrogation of samples. In the particular example chosen here, widefield images are used to record calcium activity at moderate frame rate (20 Hz here), which are then processed (within 40 ms) to take slow STED images (1 – 2.5 Hz). The key advance of the paper – which is somewhat obscured in the current manuscript – is this capability to acquire diffraction-limited resolution images at larger FOVs, rapidly identify points of interest, and then interrogate those points at higher resolution. While this capability is limited at present to 2D, and only one class of examples are given, I still think it is a useful advance for the growing field of 'smart microscopy'. Several points dampen my overall enthusiasm for the current manuscript but could probably be addressed in a revision.

Major comments

-The manuscript oversells the capabilities of the method, often claiming millisecond-scale decision making when actually the computational part of the loop is more accurately described as tens of ms. The overall time of multiscale imaging is much slower, limited by the STED imaging – which is not particularly fast over the small fields of views, as much higher frame rates (close to video rate) have been obtained by others when studying synaptic vesicle movement with STED. I would suggest the authors clarify that they are – at best – currently working in the tens of ms regime and discuss that the rate-limiting step for the overall speed is the STED imaging (and discuss how this could be accelerated). Along the same lines, ~30 nm resolution is claimed in the abstract but ~46 nm is shown on microtubules.

-Both the abstract and the discussion emphasize photobleaching and photodamage, which is odd because no photobleaching comparisons are provided, and such a comparison is somewhat meaningless because there is no advantage when imaging smaller fields of view with 'conventional' STED vs. et-STED (and it is obvious that you will have less global bleaching/damage when limiting illumination to a small rather than large FOV, holding SNR constant with a point-scanning technique). The key advance presented here is the ability to go from large to small regions of interest, and to identify those events worth going after. I suggest amending the emphasis on photobleaching to instead emphasize multiscale imaging.

-I appreciate the efforts towards a biological application, i.e., the analysis of vesicle cluster anatomy within the context of calcium events, but am unclear as to the final biological conclusion(s). Is there any dependence of clustering (or dynamics) on calcium activity, that was revealed by these measurements and not known before? I am convinced that there is a difference between event driven imaging vs. manual imaging by the MSD analysis (even though the difference appears modest), but do not understand the biological significance of these measurements. Can the authors be confident that all manual imaging was done in the absence of calcium activity, or was there simply no way to easily check without the etSTED approach? Perhaps more background would help, including more details on what is happening in the STED images of actin (Fig. 2f) and synaptotagmin (Fig. 2g) channels – I am not sure what I am supposed to appreciate. I have the same comment for Supplementary Figures 4-7 – I agree that you can STED image these structures in the context of calcium activity, but is there any dynamic event(s) in particular that matter?

-A related point: it would be useful to comment on the current pre-STED timescale of ~40 ms and the relevant biological timescale of calcium activity. What can be gained by further driving down the ~40 ms time? Please also comment on the timescale of the comparatively slow STED imaging time (hundreds of ms – seconds) and how it relates to the dynamics of the biology in question.

Other comments

-Supplementary Figure 2 will be very difficult to parse for the novice – although I believe the authors have completely specified all optics, it would be very useful to either further clarify the figure to indicate the STED, widefield, focus lock, and sample modules or provide additional figures that separately show these modules – something similar to what they have done in Fig. 1d. The lack of a substantive caption for this figure is also unhelpful. Given that the focus lock in particular is highlighted in the main text, this is particularly worth emphasizing in the figure.

-The description of the focus lock and how it works is also lacking – the few sentences provided in methods don't lend much insight. I assume this method monitors the shift in the reflected IR beam on the camera as a feedback signal, but more details or at least a reference would be helpful to those trying to implement a similar method. Also 'prolonged period of time' is vague. How long can the user do experiments before the lock must be adjusted and what sets this limit?

-I have the same comment about Supplementary Figure 1 legends – the absence of any useful description in the caption makes this figure more difficult to parse than it should be. I also strongly suggest the authors define the terms in the figure (I, th, max. filter, etc) in the caption.

-Supplementary Note 1 refers to the 'etSTED widget', and the word 'widget' is used a lot in the text. Please clarify what you mean by 'widget' – do you mean hardware? Software? Both?

The paragraph starting 'To sort out the false events...' in the Supplementary Material could benefit from a supplementary figure showing the workflow for sorting false from true events. I sort of get the idea from the description but found myself wanting a visual aid to follow along in this description.

-It would be useful to provide details on how the sub-40 ms timeline between an event -> commencing STED scales with available hardware (e.g. computer specs including GPU, CPU), and the size of the image, which will help readers further optimize and integrate the pipeline for their own applications.

-Please discuss extension of the method to 3D applications, in particular with an eye to challenges in maintaining the time scales for larger 3D stacks.

Code Review

I commend the authors for providing their code along with thorough instructions on their github site. I was impressed that they have also included a 'simulator'.

Reviewer #3:

Remarks to the Author:

The authors describe a customized plugin for ImSwitch that allows them to control their bespoke microscope system and autonomously switch between widefield and STED imaging modes when a spike in intensity in the image has been detected. This spares the sample continuous imaging under damaging STED conditions, without compromising their ability to capture the desired event with the temporal and spatial resolution needed. This method is a move forward in "smart" or "sample-adaptive" microscopy, and while this implementation works quite well for their specific use case, it is also somewhat limited to their specific use case. The widget and code they have generated should provide the building blocks for anyone wishing to do something similar, however the user will still have to optimize the code to their specific system, and the analysis/detection event they wish to perform/capture. In many cases, it might be easier to simply start over from scratch – a common problem with highly customized and individual systems. Without a demonstration of how easy it is to tailor their peak detection pipeline to a different sample/problem/microscopy system/etc., it is hard to determine how broadly applicable it is. That a microscope system could be made to switch imaging states/modes/performance based on some external input is not particularly novel, and as a tool it is not clear that the majority of users would benefit from this specific use case.

Some additional comments/questions:

- 1) This example relies heavily on the ImSwitch control software, which excludes many commercial and custom microscopy solutions – can it also be implemented in μ Manager or LabView?
- 2) How well does the peak detection pipeline extrapolate to other tissues/cells/etc.? Does it have to be "re-tuned" or calibrated for each new sample?
- 3) It is a bit unclear how false negative events were calculated, or if the authors have an estimation of how many true events were missed if those images are not saved. It is referred to in Figure 2C, but this is unclear from the figure.
- 4) It is a bit hard to work out what is going on in Fig2f and g, with the left, center, middle, and what are the arrows pointing to?

Author Rebuttal to Initial comments
--

Reviewer #1:**Remarks to the Author:**

This paper describes a very cool technique that uses automated image analysis to perform multi-scale live-cell microscopy. Specifically, the authors analyze calcium dynamics in a widefield fluorescence microscope, then zoom in on areas of interest to get super resolution STED images at areas where calcium signalling is occurring. They demonstrate this by taking super-resolution images of the cytoskeleton and a synaptic protein at synapses. It is an overall well-written paper with claims that are well-supported by their data.

The authors do a good job at describing the significance of their work, which in my estimation is: 1) They can make better use of a sample's photon budget by only doing super-resolution imaging at the areas of interest. 2) By doing so they can collect more data of rare events. This seems like it would be a very useful technique, but the application areas are far from my own expertise, so I am probably not the best judge of its potential impact on the field.

The authors also claim that their method will be applicable to other combinations of fast and high-resolution methods. I think this could use a little clarification. Their specific technique would be hard to generalize as-is, because it relies on application-specific things like the analysis of calcium signals and coordinate transformations that are (as far as I know) specific to STED. The broader concept (automated image-adaptive resolution changing or instrument control) can certainly be generalized to other types of experiments, but I don't think this is the first paper to demonstrate that concept (see comments below).

We are glad that the reviewer appreciates our technique and thankful for the informative literature. We clarify below the comments on the generalization.

The method can utilize other point-scanning methods as the high-resolution method, such as RESOLFT or confocal imaging, depending on the resolution/speed/sample type needed. It could also utilize other fast imaging methods for determining the position of interest with an online analysis, such as line-scanning, depending on the requirements of the question at hand. The method is implemented, both theoretically and the technically in ImSwitch, in a way to allow this, as it solely relies on an online-analysis of an image and transformation between imaging spaces. The coordinate transformation could be calibrated or further adjusted to any change between imaging spaces, and is thus not limited to only STED imaging. Furthermore, the analysis performed online is not limited to that of calcium signals, but rather the implementation allows the user to develop and apply the analysis pipeline of

choice.

As we show in our revised manuscript, we apply the system to not only detect calcium spikes, but also:

1. **To detect other biosensors** such as the appearance of CD63-pHluorin signal to spot exocytosis events, which was then imaged with 3DSTED.
2. **To detect protein recruitment** generating a more slowly rising signal. The dynamin signal increases as dynamin gathers to finalize endocytosis processes, such as clathrin-mediated endocytosis.
3. **To track intracellular trafficking** of endosomal vesicles and detect when they are in spatial proximity. When the distance between tracked vesicles approach the diffraction limit, STED imaging is initiated to spot potential fusion events or lipid trafficking.

These diverse applications show the versatility of the method to detect and image fast cellular processes at high spatiotemporal resolution. In fact, these analysis pipelines in the described applications have mean runtimes of 6 ms, 10 ms, 26 ms, and 61 ms (as presented in Supplementary Table 3). The STED imaging is then initiated after ~40 ms, ~40 ms, ~60 ms, and ~120 ms respectively of a detected event. These times can be further minimized using faster camera technology and higher computing power.

I see no major flaws with the their technical analysis, other than some areas described below for improving the clarity of its presentation

I hope my comments are helpful in improving the quality of the manuscript.

Sincerely,

Henry Pinkard

UC Berkeley

Specific comments

Abstract

I would remove the word “instantly” if what you really mean is less than 40ms

It might be nice to have another clause or sentence at the end saying what you used your method for and what its implications are

Thanks for the suggestions, the abstract is now updated to further reflect the additional biological applications described in the revised manuscript.

Page 1

“and tissues even dynamically” -- reads confusingly. Missing comma?

Thanks, this is now corrected.

“its most common point-scanning” -- does this mean it can be implemented in non point scanning mode? please clarify

Thanks for the chance to clarify, the sentence is now changed to include the word “single” to differ it from other scanning modes. STED imaging is most commonly implemented in single point-scanning mode. However, it can be and has been implemented in other configurations, mainly parallelized scanning by using extended light patterns or multiple donuts and excitation spots and a camera as a detector (Bingen et al 2011, Yang et al 2014, Bergermann et al 2015) where the number of scan positions are greatly reduced and higher speeds are theoretically possible, albeit the practical implementations are currently very technically limited. Additionally, line-scanning modes have been suggested and attempted (Curdts 2013, York et al 2017), again ideas that would greatly reduce the number of scan positions for an equal sized FOV. Currently, however, the best-performing STED methods are still single point-scanning modes.

It might be worth discussing adaptive approaches as a whole, not just in STED. For example: Scherf, N. & Huisken, J. The smart and gentle microscope. Nat. Biotechnol. 33, 815–818 (2015).

Li, B., Wu, C., Wang, M., Charan, K. & Xu, C. An adaptive excitation source for high-speed multiphoton microscopy. Nat. Methods 17, 163–166 (2020).

Chu, K. K., Lim, D. & Mertz, J. Two-photon microscopy with adaptive illumination power. Biomedical Optics BIOMED 2008, 1–3 (2008).

Pinkard, H., Baghdassarian, H., Mujal, A. et al. Learned adaptive multiphoton illumination microscopy for large-scale immune response imaging. Nat Commun 12, 1916 (2021).

[Disclosure: my work]

Yan X, Stuurman N, Ribeiro SA, Tanenbaum ME, Horlbeck MA, Liem CR, Jost M, Weissman JS, Vale RD. High-content imaging-based pooled CRISPR screens in mammalian cells. J Cell Biol. 2021 Feb 1;220(2):e202008158. doi: 10.1083/jcb.202008158. PMID: 33465779; PMCID: PMC7821101.

We thank the reviewer for this suggestion. In fact, we were inspired by these relevant works on smart microscopy, however the limit in number of citations forced us to prioritize the smart microscopy works related to super resolution microscopy. In light of the new applications we have now extended the manuscript to an article and thus also added a more thorough introduction referring to previous adaptive microscopy works. We hope the reviewer finds this summary and discussion more comprehensive to the field of adaptive microscopy.

Page 2

facilitate  facilitates

Thanks, this is now corrected.

flexibly optimized to the problem at hand  what does this mean? can you clarify?

Thanks for the chance to clarify. The text is now adapted to better explain the flexibility. In short: the pipelines developed, and any that can be provided by the user, can be developed to detect any event of interest as rapidly and accurately as possible. The flexibility lies in the fact that the user has the power to decide exactly what should be detected, and the optimization means that the chosen event is detected as fast and well as possible, with the preconditions that we have (computational power, algorithm availability etc.). On the contrary, a non-optimized pipeline would for example be one that detects the same events with the same accuracy, but has a longer runtime; or one that has a similar runtime but a much lower accuracy.

events such as localized calcium spikes in neurons  is this the particular event you're detecting or just one of many?

We are unsure of exactly what the reviewer means with this question, as we can see two interpretations. We here provide feedback on both our interpretations. Firstly, in line with the correction mentioned above we have now also added the additional pipelines developed to detect different types of events apart from calcium spikes in neurons. Our mentioned examples are the ones we have developed and implemented, however that is just a few out of many possibilities thanks to the general implementation of the method.

Secondly, if the reviewer is referring to the localized calcium spike events detected using the *rapid_signal_spikes* analysis. The pipeline is capable of detecting every peak present in the image, and then sorts these based on ratiometric intensity (i.e. largest ratiometric change first). This list is then filtered according to a number, which is defined by the user and inputted as a parameter of the pipeline (*num_peaks*). The selected listed events are then returned as a list of coordinates. If multiple peaks are detected, the STED imaging is then performed at the position of the brightest event detected in the ratiometric image.

Thanks to the generalized implementation of etSTED  comma other combination of dynamic events and structures can be observed -- you mean they can be used a trigger?

The dynamic events can be used as triggering events, and other nanoscale structures can be imaged with STED or other super-resolution methods. This is now also further clarified in

the text.

Furthermore, the adaptive method does not lend itself limited to use specifically  confusingly written

but instead the concept is applicable to other combinations of fast and high-resolution imaging methods.  yes, and many already have, as described above

Thank you for the comment, and as mentioned above we have now added a discussion in the early parts of the text citing the previous important works presenting automated multiscale microscopy approaches. We would like to also mention that we were aware of these works during the preparation of the manuscript, however the limit in the number of citations unfortunately forced us to prioritize other references specific to the super-resolution field.

Previous works have indeed combined mainly widefield and confocal imaging for fast and scanning imaging methods. The main improvements in the event-triggered STED imaging presented here are two-fold. First of all, **we are for the first time in a correlated and event- triggered method such as the ones mentioned by the reviewer using live super- resolution imaging as our high-resolution imaging, allowing us to probe dynamics and structural organizations on the nanoscale. Secondly, the timescales at which we are able to perform the triggered imaging is on the tens of millisecond scale, both for the time between event and STED imaging initiation as well as the frame rate of the STED imaging (as showed in the added 24 Hz imaging data on the synaptic vesicle dynamics).** These timescales are much faster than that in the previous works mentioned where the time between detected event and triggered imaging are on seconds and longer.

Two major developments enabled us to perform it on the tens-of-millisecond scale: firstly, the optimized image analysis pipelines, especially the one detecting rapid signal spikes that runs in only 6 ms, and secondly, the all-integrated control software solution we are using, which avoid delays from inter-software communication steps where data has to be shared between. This is new to this work as previous multimodal adaptive microscopy approaches did not focus on speed, since decision making on the sub-second level was not a main focus of their applications.

"an optimized analysis pipeline"  what does this mean? What would a non-optimized analysis pipeline be like??

We are sorry for the unclear language, and hope that the above clarification and changes in the main text is helpful for the reviewer to understand what we meant. In essence, the analysis methods and algorithms used are well-chosen and combined in a way to make the

runtime as short as possible while preserving a high accuracy and efficiency. For example,

for the rapid_signal_spikes pipeline detecting calcium signalling events, it runs in 6 ms while having a true event detection ratio of 91% and detecting 76% of all events.

" to confirm the validity of the event in a post-acquisition analysis "  does this mean by a human?

We have now added a reference to Supplementary Fig. 6 (reordered) at this point in the text to give an example of the type of post-acquisition analysis we meant. As of now the post-acquisition analysis is performed by the user, but one could imagine this being automated as well. See the reply three questions below for a further information on why we do it this way.

"The method can run indefinitely, and the focus lock in place in the microscope5 (Supplementary Fig. 2) maintains a stable sample throughout the experiment and ensures that the widefield and STED images are recorded in the same focal plane."  why would you need the focus lock to ensure this? Are you switching objectives?

We have now adapted the formulation in the text to clarify this point.

We are not performing objective switches, but the focus lock is pivotal to keep the sample stable over time, from seconds to hours. This ensures that the widefield imaging is always performed in the same sample plane and also that the widefield and STED imaging are always performed in the same sample plane. This can be especially handy to avoid drift when the user needs to set longer STED timelapses. While it is true that the time between the triggering widefield frame and the first STED frame is on the tens of milliseconds timescale, recording STED timelapses over a tens of seconds timescale might still see problematic axial drift without the focus lock in place.

Page 3

" By applying the analysis pipeline post-acquisition on a widefield timelapse recording (Fig. 2a) we could extract calcium intensity traces at the sites of detected events also after the detection (Fig. 2b, left). We can confirm that the detected events are, on average, in 91% (0.91 ± 0.19) of the cases true calcium spike events (Fig. 2c), as the characteristic initial spike and decay across hundreds of milliseconds can be seen"  This sounds like you're applying the same analysis algorithm offline and online. I'm guessing I'm missing something because one wouldnt expect different results doing that. Can you clarify that?

We are happy to clarify on this point. We adjusted the explanative sentence briefly to be

clearer on what we mean, but the main point is the following:

In the normal etSTED imaging, we perform widefield imaging up until the point of an event is detected, after which we immediately stop the widefield imaging and instead perform STED

imaging. After the STED imaging is finished we return to the widefield imaging. As such, for the calcium spike events and *rapid_signal_spikes* detection pipeline used, the only thing we would see in a calcium signal trace extracted from the widefield data saved from the normal etSTED imaging is the last 10 or so frames leading up to the frame that triggers the event, i.e. the sharp increase. When we later return to the widefield imaging that calcium spike has already passed, and as such we cannot follow the ~seconds decay of the calcium signal.

Instead here we have acquired uninterrupted widefield timelapses, without triggering STED imaging, in order to record the full calcium events. Then, we can run the same analysis as we would online during normal etSTED imaging, and see what frames and locations triggers event detection. Then, thanks to having the full widefield timelapse we can extract the calcium intensity traces both before (as in normal STED imaging) and after the spike events that would have triggered STED imaging. Using the extracted calcium signal traces, we can more surely confirm that they are indeed real calcium events.

" Important to note is that these events can be sorted out in post-acquisition analysis by inspecting the calcium trace up to the point of event detection " Why wouldnt you just use this info in real time if you have it?

Again, thanks for giving us the chance to clarify this part if it was not clear. First of all, we would like to refer to the end of Supplementary Note 7 (reordered) where the post-acquisition true/false event decision is described, and this suggestion has been discussed.

The reason why it is not included in the real-time analysis and decision making is that that extraction and analysis of the calcium traces would slow down the real-time analysis pipeline significantly. As previously discussed, there is often a direct trade-off between runtime and accuracy, and having a 100% accurate pipeline would mean a significantly slower pipeline and thus the loss of possibility of the method to investigate what happens during the hundreds of milliseconds following an event. As we want to be as fast as possible at the triggered site with our STED imaging we chose to perform this double check afterwards and simply discard the few events that turn out to be falsely detected events. The benefit is a faster time between detected event and STED imaging, at the cost of an occasional impact on the sample of a few extra recorded STED imaging.

One could imagine a future development of the method where a pipeline, if necessary, is further supplemented with a validity check pipeline. This could automatically run after the STED imaging at the event has been performed, and decide whether the data should be

saved or not. However, the capability of the current implementation is not affected by the lack of such an auxiliary check.

Page 5

"inside < 40 ms"  "with 40 ms"

"proving the potential for correlating millisecond dynamics such as calcium sensing in living cells with super-resolved details of proteins of interest."  maybe make this more clear that its just referring to the data collection itself, not analysis you did on it

Thanks, these comments have now been implemented in the text for correction and clarification.

" in the order of 100–1000x faster" -- citation or calculation supporting this?

We have now provided a calculation supporting this at the statement in the text. The number comes from a calculation of the recording time of a single frame (frame period) in the 80x80 μm^2 full-FOV timelapse, as compared to that of a smaller FOV on the order of single micrometres across. This is the time difference spent on a single frame during investigation of the sample area (monitoring it with the widefield in the etSTED experiments, and only triggering STED imaging during an event of interest). The sheer difference in size between the two means that a speed-up of >1000x is achievable, depending on what size one chooses of the triggered STED image. In the added synaptotagmin data recorded in a 1x1 μm^2 region of interest we reach frame periods of 41 ms, thereby accelerating the recording by 5000x.

Scan size 80x80 μm^2	Frame period = 218.1 s	
Scan size 3x3 μm^2	Frame period = 336 ms	→ 650x faster
Scan size 2x2 μm^2	Frame period = 153 ms	→ 1430x faster
Scan size 1x1 μm^2	Frame period = 41.3 ms	→ 5270x faster

"Moreover, it allows experiments not previously feasible, where the dynamics and structure of synaptic vesicle pools can be directly correlated to the presence or lack of local calcium oscillations" -- might be a good place to talk about biases your method could introduce

Here we cannot think of specific bias that might be introduced other than detecting false events when the SNR is too low. These false detections can, as mentioned, be sorted out in post-acquisition analysis using the widefield images saved from each event, and were well characterized in Supplementary Fig. 6 (reordered).

"while the method as proposed here combines widefield imaging of a calcium sensor with STED imaging of actin, microtubules, or synaptic vesicles, the possibilities to generalize and extend the method are endless." -- I think this is not the first and only implementation of this general idea, and so others should be cited

Thanks again for noting this, as mentioned above we have adapted the manuscript to include a fairer discussion on previous adaptive microscopy works, and why our developed method presents an edge as compared to those.

"to the accuracy and speed of liking." Not exactly true. Some of the analysis routines you described just above this might take a nontrivial amount of computation time

We thank the reviewer for the comment, and would like to clarify what we mean, both in the text and here.

First of all, in the updated manuscript and code we are providing three varying detection pipelines, all with average runtimes in the < 100 ms regime. We show that even a more complex detection pipeline such as tracking vesicles and detecting proximity of vesicles, with information from up to 15 widefield frames at each step, can have a runtime in the < 100 ms regime during real-time acquisition. While this might be considered a nontrivial amount of time, it is still fast enough to capture the interaction of said vesicles in the STED imaging and correlating the information. Additionally, we envision this pipeline to run even faster by implementing a GPU-accelerated track connection step as well, whereas that step currently runs on the CPU. Simpler pipelines, such as `rapid_signal_spikes`, "only" compares two frames against each other and is thus bound to be faster.

What we mean with "to the accuracy and speed of liking" more has to do with the balance between the two. As mentioned earlier, the runtime and the accuracy of a pipeline are often highly connected, and if we want to develop a more accurate pipeline the runtime will often increase, unless specific tricks or faster algorithms are used. Thus, we can develop a pipeline to be extremely accurate, but with a longer runtime, if accuracy is our biggest concern and 6 or 100 ms runtime does not matter for our application. Instead, if for the application it is crucial to be very fast, we might sacrifice a bit of accuracy at the price of sorting through more data in the data analysis phase.

Figure 1

You shouldn't use red and green to differentiate things, as they appear identical for the most common type of color blindness. For a great discussion of this see:

<https://www.nature.com/articles/s41467-020-19160-7>

b) "Calc" is not described in the caption

c) Those inset images are incredibly tiny, and I'm not sure what I'm supposed to take away from

them

d) Lots of acronyms on this that aren't defined anywhere. Also, theres a lot of detail that I think is not releveant to the main point of the paper. Maybe it makes sense to have this as a

supp figure, where it can be fully explained, and have a simplified block diagram in this figure e/f) These seem to be technical details that could perhaps be moved to the supplement. It seems to me that the coordinate shift is more of an implementation detail than a core part of the method. I don't understand why you need to prove why the STED resolution would be affected by the triggering method. Also, there's no control to compare to here. Also, this is really the width of a microtubule imaged under STED, not the actual resolution, right?

We thank the reviewer for the input on figure 1. We have now adapted the figure to mainly a graphical representation of the method as a whole that we believe is easier for interpretation and overview of the method. The microscope, now in b), is changed to a more simplified block diagram, as the suggestion. The more technical details of the microscope are all found in Supplementary Fig. 3.

Regarding the red/green colours to differentiate things, we fully agree with the reviewer and thank you for making us observant of what we had missed, supposing the reviewer meant panel a). We have corrected this here and in the other figures to colours that should be well separable for colour-blind readers.

Regarding the STED resolution, it is correctly as you mention the width of an imaged microtubule that we report. We have now adjusted the text segment to better reflect this. In essence, we can estimate our spatial resolution to be around 30 nm from these images of the microtubules, knowing the physical size of the microtubules and labelling molecules themselves. Please see a further comment on this below in the reply to a question from reviewer #2 regarding the same point.

We thank also the suggestion from the reviewer regarding moving some of the details to the Supplementary Information. The quantification of the size of STED-imaged microtubules, together with the example experiments using the *rapid_signal_spikes* pipeline to detect calcium activity in HeLa cells, is now moved to Supplementary Fig. 5. With regards to the residual shift between widefield event detection coordinate and scanned STED centre we have also moved this to Supplementary Fig. 4. While it is central to have a functioning coordinate transformation for the method to work accurately, we agree that for readability this fits better in the supplementary. We hope that the overall rearrangement and simplification of the figure makes it more readable for the reader.

Figure 2

There is a ton of information in here, and a lot of text packed in. I think it could benefit a lot from trimming down to what you think are the most important takeaways for the reader, and moving other stuff to supplement. It also seems like there are several places where you have

abbreviations that you dont define in the caption, which makes the figure a lot harder to understand

We thank the reviewer for the suggestion, we now reorganized Figure 2, by moving the etSTED imaging of actin to Supplementary Fig. 8, as well as added a schematic panel explaining the question and parameters for the etSTED method for the etSTED imaging of synaptic vesicles. We have further clarified the abbreviations in the legend. We hope that these changes will make it easier for the reader to follow. The current form of the figure keeps the crucial information for the presentation of the method and application to calcium sensing as an event of interest.

b) Define dF/F. State what annotated event coordinates are in the caption (in addition to text)

Thanks for pointing out these mistakes, we have now amended the figure caption to include this.

c) I'm confused as to why this is a distribution, rather than just a number with some error bars. My understanding is that this is meant to quantify the accuracy of your detection algorithm. Would something like an ROC curve or confusion matrix be a better description of this?

We hope the following explanation is informative for the reviewer: We have quantified the ratio of true positive event detections (True) as well as the ratio of detected real events (Det) as a ratio of events. This quantification is done in multiple experiments and multiple cells, as it can depend on the background in the cell, the structure of the cell, the labelling density, as well as the acquisition parameters together with the user-inputted parameters of the detection pipeline. What is plotted is one data point for each cell in the various experiments analysed.

d) Again, don't use red/green

Thanks again for making us notice, this is now corrected with a lighter hue of red that should be distinguishable for people with deuteranopia.

f) Overlapping text on leftmost picture makes it hard to read. I dont think you need "Rat zoom". You could make arrows going to where it came from in larger FOV. You dont describe what the white outlines are in the caption

We thank the reviewer for the suggestions. There are now lines connecting the ratiometric zooms from the maximum projected ratiometric images. "Rat. zoom" is now also removed.

Regarding the white outlines, they are actually already explained in the caption: “*Overlaid*

semi-transparent white outline shows extent of detected local calcium activity". We have also performed other small adjustment to improve readability of the figure.

Reviewer #2:**Remarks to the Author:**

In this manuscript, Alvelid and coworkers describe a framework that allows multiscale optical interrogation of samples. In the particular example chosen here, widefield images are used to record calcium activity at moderate frame rate (20 Hz here), which are then processed (within 40 ms) to take slow STED images (1 – 2.5 Hz). The key advance of the paper – which is somewhat obscured in the current manuscript – is this capability to acquire diffraction-limited resolution images at larger FOVs, rapidly identify points of interest, and then interrogate those points at higher resolution. While this capability is limited at present to 2D, and only one class of examples are given, I still think it is a useful advance for the growing field of ‘smart microscopy’. Several points dampen my overall enthusiasm for the current manuscript but could probably be addressed in a revision.

We thank the reviewer for the comments and suggestions. Please see the point-by-point replies below regarding all the additions and changes we have made to the manuscript in order to clarify and the new experiments we have performed in order to prove the strengths of the method.

Major comments

-The manuscript oversells the capabilities of the method, often claiming millisecond-scale decision making when actually the computational part of the loop is more accurately described as tens of ms.

We thank the reviewer for the comments on this, and we have adapted the mentions of “millisecond timescale” to “tens-of-milliseconds timescale” to avoid confusion for the reader.

The overall time of multiscale imaging is much slower, limited by the STED imaging – which is not particularly fast over the small fields of views, as much higher frame rates (close to video rate) have been obtained by others when studying synaptic vesicle movement with STED.

We thank the reviewer for the comment and would like to point to the new experiments added in the manuscript showing STED imaging of synaptic vesicles at 24 Hz. The previously shown 2.5 Hz imaging of the synaptic vesicles was limited both by the size of the

FOV ($3 \times 3 \mu\text{m}^2$) and additionally by software design choices affecting the time between recorded frames. Decreasing the imaged FOV to $1 \times 1 \mu\text{m}^2$ and completely removing the software limitations we can now perform close to video rate STED imaging, only limited by the physical trade-off of point-scanning techniques between size of FOV and frame rate.

Further decreasing the dwell time (now 30 μ s) as well as the imaged FOV would push the imaging towards higher frame rates if needed for answering a specific biological question.

I would suggest the authors clarify that they are – at best – currently working in the tens of ms regime and discuss that the rate-limiting step for the overall speed is the STED imaging (and discuss how this could be accelerated).

We thank the reviewer for the suggestion, and we are now providing a clarification and short discussion regarding the rate-limiting steps in the method, at the end of the main text.

Along the same lines, ~30 nm resolution is claimed in the abstract but ~46 nm is shown on microtubules.

We thank the reviewer for this comment, and would like to clarify the difference between observed width of a biological structure in a fluorescence microscopy image and optical resolution or size of PSF. We have further amended the text segment mentioning this as well, to better reflect what we mean.

The spatial resolution, or size of the effective STED-PSF, is defined by how efficiently we deplete the fluorescence in the periphery of the focal spot, which is strictly related to the properties of the fluorophores, the quality of the light pattern and the temporal synchronization of the illumination. It is however not affected by the underlying biological structure we are labelling. The resulting STED image can be well described by the convolution between a STED-PSF of a certain width and the structure. The microtubules are extended biological structures with an outer diameter of 25 nm as provided by EM data. If we consider the size of the label Sir-Tub in this case, we should add additional 2 nm. Thus, simulating the convolution process and assuming for simplicity that the microtubule is a step function and the STED-PSF is about 30 nm, the microtubules width in the final images will be roughly ~40 nm. In the imaging of the microtubules we were additionally using a pixel size of 25 nm, same as the 25-30 nm used in the etSTED imaging of the dynamic processes in order to optimize STED imaging frame rate and timelapse length, not allowing us to resolve structures below 50 nm according to the Nyquist limit. With these two aspects in mind, one could expect images with an average microtubule fitted width of around 45 nm, agreeing well with our observations.

-Both the abstract and the discussion emphasize photobleaching and photodamage, which is odd because no photobleaching comparisons are provided, and such a comparison is somewhat meaningless because there is no advantage when imaging smaller fields of view

with 'conventional' STED vs. et-STED (and it is obvious that you will have less global bleaching/damage when limiting illumination to a small rather than large FOV, holding SNR

constant with a point-scanning technique). The key advance presented here is the ability to go from large to small regions of interest, and to identify those events worth going after. I suggest amending the emphasis on photobleaching to instead emphasize multiscale imaging.

We thank the reviewer for the comment and would like to argue our case and the reason for our emphasis here. The benefits provided from the method is threefold in our view. Firstly, the sample is exposed to STED illumination only around the POI (point of interest) in a smaller FOV, meaning that certain parts of the cells will never be exposed to STED since they did not show any events of interest. This is not the case for conventional raster scanning STED where all the FOV is probed with STED intensity. Secondly, the POI is exposed to STED intensities only for a specific temporal window at the moment of the detected event.

These two points overall limits photobleaching and photodamage compared to conventional STED approaches as the photon budget from the fluorophores is used in the exact moment and position of interest. This is true not only globally for the whole cell, but also locally, as you will not perform STED imaging with the etSTED method unless there is an event of interest occurring. Thus, during an experiment, you can record the same amount of information content as for a large FOV STED timelapse covering the whole $80 \times 80 \mu\text{m}^2$ region investigated with a lower photobleaching and photodamage across the sample and using the available photon budget in a more efficient way.

The third benefit is the multiscale approach as appreciated by the reviewer; where fast dynamic events on a larger scale, using the speed, photon budget, and gentleness of widefield imaging are correlated with super-resolution imaging with very high spatiotemporal resolution in a small region, in a way that previously has not been shown. This really pushes STED imaging to its full spatiotemporal potential without wasting precious photons and cycling in region and moments not relevant for specific dynamic studies.

Previous multiscale approaches used for example widefield and confocal, but never super-resolved imaging such as STED. Additionally, to our knowledge, the analysis, triggering, and shift in imaging methods took seconds or even minutes. Here we triggered STED imaging for the first time, and we optimized the software and analysis pipeline to be as fast as possible, i.e. tens of milliseconds, which is now something we reinforced in the main text.

-I appreciate the efforts towards a biological application, i.e., the analysis of vesicle cluster anatomy within the context of calcium events, but am unclear as to the final biological

conclusion(s). Is there any dependence of clustering (or dynamics) on calcium activity, that was revealed by these measurements and not known before? I am convinced that there is a difference between event driven imaging vs. manual imaging by the MSD analysis (even

though the difference appears modest), but do not understand the biological significance of these measurements.

We thank the reviewer for the comments and we clarify the biological context of our observation as follows.

In the presynaptic active zone, the transient increase in intracellular calcium leads first to synaptic vesicle exocytosis from the pool of vesicles docked at the presynaptic active zone (also known as the readily releasable pool), within milliseconds after stimulation. This event does not lead to profound changes in the positioning of the fluorescently-labelled synaptotagmin, since these molecules are displaced during exocytosis by, at most, ~20-30 nm. Exocytosis is then followed by two events, which take place over a few tens of milliseconds:

First, the fused vesicles diffuse laterally, away from the active zone, to the peri-active zone area, where they can be later endocytosed. This diffusion behaviour will affect only the fused vesicles, and will only result in a lateral displacement of ~50-100 nm, to the “edges” of the active zone area.

Second, the free slots on the active zone are rapidly filled by recycling pool vesicles that reach the active zone by diffusion, and not by directed motion. For reviews on vesicle pool behaviour, please see Rizzoli and Betz, 2005; Rizzoli, 2014. The mobility of the recycling pool vesicles is not enhanced by stimulation, as they seem to move at the maximum possible velocity at all times (see for example Gaffield and Betz, 2007; Westphal et al., 2008; Kamin et al., 2010). Since most of the labelled vesicles find themselves in the recycling pool under our imaging conditions, it is clear that stimulation can only result in a limited difference in the presence of activity (“modest”, as noted by the Reviewer).

These initial events take place, as mentioned above, within a few tens of milliseconds (see reviews by Haucke et al. 2011, or Rizzoli 2014), and are probably finished by the time we initialize our measurements. **In Alvelid et al., we use etSTED to image the dynamics of synaptic vesicles recycling pool upon Ca²⁺ sensing in the time domain of 50 msec to 10 sec. This time domain includes the following:**

First, the fused synaptic vesicles induce an imbalance in the local plasma membrane tension, which in turn leads to the endocytosis of vesicle material from the plasma membrane.

Second, endocytosis takes place in the peri-active zone area, and may involve different mechanisms (see review by Gan and Watanabe, 2018). Without discussing these mechanisms in detail, it is important to point out that they all share one common aspect,

namely that the synaptic vesicles are propelled towards the inner side of the synapse, most often by the involvement of actin polymerization (already described almost two decades ago, Bloom et al., 2003 1). This results in a specific and directed displacement of the vesicles undergoing endocytosis, **which is the only event that is predicted to induce a specific change in MSD under our experimental conditions.** The endocytosis event is quickly followed by actin de-polymerization, and these vesicles join the diffusing recycling pool over the next seconds.

We would like to point out that our activity conditions imply that the neurons undergo rare bursts of activity, of up to ~80 action potentials (Truckenbrodt et al., 2018), at irregular intervals. These short bursts are unable to affect the so-called reserve pool (~50% of the vesicles). Longer stimulation, or the application of strong stimuli as KCl-induced depolarization (Joensuu et al., 2) affect this pool, inducing its mobilization, but this is not a physiologically-relevant phenomenon under our imaging conditions.

Overall, **our technique, unlike past works, enables us to monitor the dynamics of synaptic vesicles responding specifically to the normal neuronal activity, free of non- physiological external stimuli.** Our experiments reveal especially the endocytosis step that takes place after stimulation, and thus confirm previous models of synaptic vesicle behaviour (e.g. Haucke et al., 2011; Rizzoli, 2014), while excluding hypotheses on the directed motion of synaptic vesicles to active zones, or on the strict localization of vesicle pools in specific areas, which have long been present in the literature.

Can the authors be confident that all manual imaging was done in the absence of calcium activity, or was there simply no way to easily check without the etSTED approach? Perhaps more background would help, including more details on what is happening in the STED images of actin (Fig. 2f) and synaptotagmin (Fig. 2g) channels – I am not sure what I am supposed to appreciate. I have the same comment for Supplementary Figures 4-7 – I agree that you can STED image these structures in the context of calcium activity, but is there any dynamic event(s) in particular that matter?

As indicated above, manual imaging of synapses will result in data reporting mostly the diffusive behavior of the recycling pool vesicles, which represent virtually all of the labelled structures, and which are not modulated by activity under physiological conditions. It is only by etSTED that we can report the active displacement that takes place after stimulation, and which is most likely related to endocytosis, as mentioned above. In other words, **etSTED is able to reveal, for the first time, rare activity events that would otherwise be lost in the noise produced by diffusive motion.** Future works, in which other labels will also be

included, as active zone markers or membrane labels, will enable us to perfect this analysis,

and to derive more detailed information on the poorly understood series of events that leads from exocytosis to endocytosis.

Relating to actin, we would like to point out that, while filamentous actin is involved into the recycling of synaptic vesicles, as mentioned above, the overwhelming majority of actin molecules in the presynaptic bouton are involved in the structural maintenance of the synapse. We therefore did not perform further dynamic quantification. An in-depth characterization with different actin binding molecules would allow to assess whether polymerization or depolymerization of f-actin occurs within 10 s since calcium sensing, as expected, but this is beyond the scope of the current manuscript.

-A related point: it would be useful to comment on the current pre-STED timescale of ~40 ms and the relevant biological timescale of calcium activity. What can be gained by further driving down the ~40 ms time? Please also comment on the timescale of the comparatively slow STED imaging time (hundreds of ms – seconds) and how it relates to the dynamics of the biology in question.

Having the ability to go down to 1-2 ms after the calcium event would allow us to catch synaptic vesicles exocytosis. Unfortunately, this is not yet possible, due to the recording time of available camera technology and the time required for the analysis pipeline to run, but should be an achievable target in the near future. Nevertheless, **in our most recent datasets we pushed the temporal resolution by acquiring STED timelapses at 24 Hz to demonstrate that etSTED is not technically limited to 2.5 Hz.** Ultimately, as in previous STED works on synaptic vesicles, the recording time scales with the sampling frequency and the pixel dwell time, which depend on the labeling densities. As the labeling is comparable to previous studies, we could achieve similar speeds.

The faster framerate in principle allowed to resolve more frequently single moving vesicles and to monitor their dynamics in time domain related to endocytosis mechanisms. In the added Supplementary Fig. 13, we performed STED imaging at 24 Hz and the use of different labelling densities allowed us to resolve single synaptic vesicles (60 nm FWHM) (Supplementary Fig. 13a) and their directional movements, color-coded from blue to red, as shown for two representative synaptic boutons (Supplementary Fig. 13b-c).

1. Bloom, O. et al. Colocalization of synapsin and actin during synaptic vesicle recycling. *J Cell Biol* **161**, 737-747 (2003).
2. Joensuu, M. et al. Subdiffractional tracking of internalized molecules reveals heterogeneous motion states of synaptic vesicles. *J Cell Biol* **215**, 277-292 (2016).

Other comments

-Supplementary Figure 2 will be very difficult to parse for the novice – although I believe the

authors have completely specified all optics, it would be very useful to either further clarify the figure to indicate the STED, widefield, focus lock, and sample modules or provide additional figures that separately show these modules – something similar to what they have done in Fig. 1d. The lack of a substantive caption for this figure is also unhelpful. Given that the focus lock in particular is highlighted in the main text, this is particularly worth emphasizing in the figure.

We thank the reviewer for this comment and chance to clarify. According also to the comments of reviewer #1 we have now adapted and simplified the setup in Fig. 1d to a boxed sketch of the important modules of the setup. We also would like to emphasize that Supplementary Fig. 3 (reordered) is a complete schematic view of the setup for the more optically-inclined reader, but agree that more clarity would be useful also for that case. According to the suggestions we have now extended the caption and added module boxes around the individual modules of the setup. Regarding the focus lock, see the reply below.

-The description of the focus lock and how it works is also lacking – the few sentences provided in methods don't lend much insight. I assume this method monitors the shift in the reflected IR beam on the camera as a feedback signal, but more details or at least a reference would be helpful to those trying to implement a similar method. Also 'prolonged period of time' is vague. How long can the user do experiments before the lock must be adjusted and what sets this limit?

We are sorry for the confusion on this point, but would like to clarify. The focus lock is highlighted in the text as it was important for performing the experiments, but not explained in more detail in this manuscript as it is previously published in an earlier work (Alvelid et al. 2020). That article is also cited in the main text when the focus lock is mentioned. The focus lock is explained in good detail with an extra schematic panel showing the functionality of it in that article, where we also provide the software part available in GitHub for easy implementation in other systems.

In order to be more clear the article is now also cited where the focus lock is mentioned in the Methods section.

For the time it can run without manual intervention it is at least multiple hours. As can be seen in the cited article, large-scale recordings with >4 hours of acquisition time was performed without any issues. Longer periods of time than that has not been tested but is expected to not be a problem. We adjusted our formulation in the Methods now to be more

precise.

-I have the same comment about Supplementary Figure 1 legends – the absence of any useful description in the caption makes this figure more difficult to parse than it should be. I also strongly suggest the authors define the terms in the figure (l, th, max. filter, etc) in the caption.

We have now provided a more thorough figure legend, explaining also the abbreviations in the figure. We further refer to Supplementary Note 1 where the analysis pipeline is described in great detail.

-Supplementary Note 1 refers to the 'etSTED widget', and the word 'widget' is used a lot in the text. Please clarify what you mean by 'widget' – do you mean hardware? Software? Both?

We are sorry if this was not clear in the Methods and supplementary notes, and have now clarified this in the Methods section under both Microscope control and etSTED widget.

With widget we mean a module of a software, in this case the microscope control software, that is developed to control a specific part of the microscope. ImSwitch, the microscope control software used to implement the method in the manuscript, is built in a modular fashion with software widgets controlling separate pieces of hardware or parts of the microscope, such as a laser widget, scanning widget, detector widget, etc. These widgets are interacting, more or less, with each other, to give the unified control of the microscope. As such, the etSTED widget developed to control the etSTED method is performing the behind-the-scenes control of running the analysis pipelines, performing the coordinate transformation, etc., and is interacting with other widgets of the control software for the image acquisition, laser control, and scanning for example.

The paragraph starting 'To sort out the false events...' in the Supplementary Material could benefit from a supplementary figure showing the workflow for sorting false from true events. I sort of get the idea from the description but found myself wanting a visual aid to follow along in this description.

Excuse us for forgetting to refer to Supplementary Fig. 3 (now 6) in Supplementary Note 2 (now 7), which is exactly what you are seeking. Thanks for noticing, it is now amended.

-It would be useful to provide details on how the sub-40 ms timeline between an event -> commencing STED scales with available hardware (e.g. computer specs including GPU, CPU), and the size of the image, which will help readers further optimize and integrate the

pipeline for their own applications.

We thank the reviewer for the suggestion that we agree would be a great resource for readers interesting in implementing the method. We have characterized the temporal performance of the method, and especially the runtime of the analysis pipeline, as fully presented in the added Supplementary Note 8, Supplementary Fig. 7, and Supplementary Table 3.

The runtime of the analysis pipeline is characterized for all three developed pipelines, and characterized for the processing unit used (GPU-accelerated or CPU only) as well as the size of the widefield image. As expected, the runtime of the pipelines decreases with decreasing widefield image size, down to only 2 ms for the *rapid_signal_spikes* analysis pipeline with widefield images of 200 x 200 pixels. Unfortunately, we do not have the possibility to perform the characterization on different computers with different hardware specifications, but the results are expected to scale positively with the available computing power, especially the GPU-accelerated performance of the analysis pipelines.

The pipeline runtimes are reported from testing on a test sample with fluorescent beads, for fair comparisons between different widefield image sizes as well as CPU and GPU. Additionally, pipeline runtimes from real etSTED experiments in HeLa cells and neurons with 800x800 pixels widefield images are reported, for a more accurate estimate of runtimes in real samples.

Additionally, in Supplementary Fig. 7, for experiments with the *rapid_signal_spikes* pipeline, we present the times from the start of a widefield exposure to the start of STED scanning when an event is detected, from the test experiments. This is the pipeline runtime with added widefield exposure time (20 ms) and the so called overhead time (coordinate transform, scan curve calculation, laser toggling, other software executions), as previously mentioned in the main text, and shows that the overhead time is not dependent on the pipeline runtime or other imaging conditions, as expected since they are separate. Furthermore, the overhead time is not dependent on which pipeline is used.

-Please discuss extension of the method to 3D applications, in particular with an eye to challenges in maintaining the time scales for larger 3D stacks.

We thank the reviewer for this comment. We now provided new experiments where etSTED was used to trigger 3DSTED imaging with additionally improved axial resolution using a combination of the donut and the so called top-hat phase plate.

As shown in the new experiments where we imaged exocytosis upon pH sensing or

endocytosis upon dynamic recruitment of a specific protein, the method is not limited to neither 2D STED nor XY imaging. We show 3D STED imaging in XZ cross-sections across

our detected events of interest. In such recordings the temporal resolution is not hindered in any more way than that of 2D STED imaging in XY, and indeed we show imaging at up to 11 Hz in XZ, that could further be pushed towards higher temporal resolution by decreasing the image size, just as in the new experiments added with 24 Hz STED imaging of the synaptic vesicle dynamics.

Extending the method to 3D imaging could include volumetric STED imaging and/or 3D localization of events of interest. While the 3D STED imaging in XZ cross-sections of detected potential exocytosis/endocytosis events that we performed was limited to imaging the plasma membrane, enabled by keeping the plasma membrane closest to the cover glass surface in focus, performing localization and volumetric STED imaging of an event taking place anywhere in the cellular volume poses additional challenges. 3D localization of events would likely require further image recording and processing for detecting the event, and as such would definitely require a longer detection time from exposure to STED scan start. The lack of optical sectioning in widefield imaging is further complicating the picture, and likely other fast methods or complementary read-outs would be needed for such a development.

When it comes to volumetric STED imaging there is nothing technically limiting it and the method is capable of triggering any kind of STED imaging at the moment. The challenge is, as the reviewer mentions, to maintain a high temporal resolution when performing volumetric imaging. This is not a unique challenge to this method, but instead is a problem that applies to any point-scanning microscopy. As a back-of-the-envelope calculation, based on the data presented in the manuscript, a 2D STED image of $1 \times 1 \mu\text{m}^2$ takes 42 ms to acquire. As such, a $1 \times 1 \times 1 \mu\text{m}^3$ volume (with a Z step size of 60 nm) would take $42 \text{ ms} * (1 \mu\text{m} / 0.060 \mu\text{m}) = 714 \text{ ms}$ to acquire. Thus, we could reach volumetric imaging rates of $\sim 1 \text{ Hz}$ following a triggering event. This is of course many orders of magnitude faster than if you are performing traditional volumetric 3D STED imaging of large fields of view, and as such this method is also capable of improving that aspect, even more favourably than for the 2D imaging presented due to the extra dimension. While possible, there are two challenges with using volumetric 3D STED imaging in these kinds of experiments: the temporal resolution of $\sim 1 \text{ Hz}$ is likely not high enough to follow many of the dynamic processes we are interested in, but most of all the repetitive exposure of illumination light of the imaged planes would cause a faster photobleaching if we count the number of available imaged time points as compared to 2D STED imaging. The length of the timelapses would likely be another limitation to what processes one could follow with such an imaging. However, as mentioned, this is not a challenge unique to the event-triggered STED method, and indeed the method can still

improve on both the temporal resolution of volumetric imaging as well as the photobleaching as compared to traditional volumetric STED imaging of larger fields of view.

Code Review

I commend the authors for providing their code along with thorough instructions on their github site. I was impressed that they have also included a 'simulator'.

We thank the reviewer for the kind words, and would also like to mention that the widget is now provided in the main version of ImSwitch (kasasxav/ImSwitch), featuring a full simulation mode, as well as a standalone widget (jonatanalvelid/etSTED-widget) with a simulated camera and image viewer to perform exactly what was possible in the previously referred ImSwitch-etSTED repository (jonatanalvelid/ImSwitch-etSTED).

Reviewer #3:**Remarks to the Author:**

The authors describe a customized plugin for ImSwitch that allows them to control their bespoke microscope system and autonomously switch between widefield and STED imaging modes when a spike in intensity in the image has been detected. This spares the sample continuous imaging under damaging STED conditions, without compromising their ability to capture the desired event with the temporal and spatial resolution needed. This method is a move forward in “smart” or “sample-adaptive” microscopy, and while this implementation works quite well for their specific use case, it is also somewhat limited to their specific use case. The widget and code they have generated should provide the building blocks for anyone wishing to do something similar, however the user will still have to optimize the code to their specific system, and the analysis/detection event they wish to perform/capture. In many cases, it might be easier to simply start over from scratch – a common problem with highly customized and individual systems.

Without a demonstration of how easy it is to tailor their peak detection pipeline to a different sample/problem/microscopy system/etc., it is hard to determine how broadly applicable it is.

We thank the reviewer for the critical feedback, and would like to point to the experiments and new data added in Figure 3, Figure 4, and Supplementary Fig. 14–16. This data and the text discussing it shows that both tailoring a specific pipeline, such as `rapid_signal_spikes`, to detected from a detection-point-of-view similar types of events, as well as implementing various analysis pipelines, to detect widely different events, is entirely possible. Different samples can be used, different questions can be investigated, and different microscopy systems could additionally be used. In this way, we hope it is now clearer for the reviewer and the reader that the method is broadly applicable.

That a microscope system could be made to switch imaging states/modes/performance based on some external input is not particularly novel, and as a tool it is not clear that the majority of users would benefit from this specific use case.

We agree with the reviewer that multiscale imaging approaches has been previously presented, but those never included live super-resolution imaging and especially STED. A second novel aspect comes from the speed at which this can be achieved, which is crucial for the cellular processes we observed. Please see the replies to reviewer #1 for a more in-

depth discussion on this. We also added additional references in the extended discussion on the subject in the text.

Some additional comments/questions:

1) This example relies heavily on the ImSwitch control software, which excludes many commercial and custom microscopy solutions – can it also be implemented in μ Manager or LabView?

We thank the reviewer for a chance to explain the software choices further. ImSwitch was developed as a microscope control software in the recent years in order to be a general microscope control software solution. To this end, we have also developed it to be applicable to any point-scanning technique, by implementing support for general scanning and point-detectors using NiDAQ devices. This is a support that is lacking in the otherwise powerful μ Manager for example, that to date does not have a standard implementation or plugin for laser scanning. To our knowledge, stage scanning applications have been developed, but laser scanning is still missing. There is one project called OpenScan (<https://eliceirilab.org/openscan/>) that aims at solving this, however it seems as if this project has been put on hold. Additionally, support for multiple detectors, even multiple cameras, is limited in μ Manager, where from our experience even running two cameras with different image sizes is not possible. As such, μ Manager in its current form with current plugins is not a solution that is feasible for controlling an event-triggered acquisition method.

Furthermore, the recent python-based control software/packages pyMMcore and pycromanager (Pinkard et al. 2021) are software/packages that could be an alternative for python-based microscope control, however as they still are based on and interface the μ Manager core libraries that are used for device control and as an acquisition engine, the limits mentioned above also applies to pycromanager. In the extensive documentation for pycromanager one laser-scanning application for second harmonic generation microscopy is mentioned, however this uses the still-not-available OpenScan project.

With regards to LabView, it is another powerful programming approach allowing implementation of microscope control software. Implementing a similar and fast event-triggered acquisition method is likely possible in LabView, however that implementation would unfortunately have to be made from scratch, following our template provided in Supplementary Note 5, as LabView follows an entirely different programming architecture as compared to Python or other object-oriented programming languages. Additionally, the analysis pipelines would have to be translated to LabView graphical programming code, and as such there is little code that could be reused in this case.

Nevertheless, the provided generalized widget can be a direct implementation for microscope users using Python-based microscope control software, where the widget could be implemented by connecting a few signals to basic hardware control specific to their

control software. Additionally, we envision that the provided generalized widget could be used as an external and standalone widget that could connect to other microscope control software using intersoftware communication channels, although limiting the speed due to intersoftware communication and data transfer.

It is also important to note that the generalized and modular software structure of ImSwitch is one of the driving factors that allowed this method to be implemented and performing as fast as it is. The single-software solution for controlling multiple different acquisition methods such as widefield and point-scanning allows an implementation of a fast control without intra-software communication as in most previous adaptive microscopy methods. While solutions for adaptive microscopy earlier published such as AutoScanJ (Tosi et al 2021) aims at providing generalized solution able to interact with various control software, even commercial, it necessarily comes at a lack of speed due to the inter-software communication and data transfer. In their work, the time resolution between event detection and higher-resolution imaging (confocal) is not discussed. This is a point where our solution provides an edge, allowing correlation of events and dynamic investigations with STED on the tens-of-millisecond scale.

Lastly, when it comes to commercial microscope control software, these are often black boxes without any possibility to externally modify or control them. Thus, implementing this rapid control method in a commercial microscope is unfortunately not currently possible. This is instead the strength of using custom-written and open-source microscope control software: it allows development of acquisition methods and automation in a way that is not possible for the user on a commercial microscope. That being said, there are no technical limitations in commercial microscopes and control software limiting the implementation of similar, rapid event-triggered acquisition methods, and we envision this happening in the near future.

2) How well does the peak detection pipeline extrapolate to other tissues/cells/etc.? Does it have to be “re-tuned” or calibrated for each new sample?

The peak detection analysis pipeline *rapid_signal_spikes* works well to detect various events in various cell types, such as the examples shown in both hippocampal neurons and HeLa cells for calcium signalling events and CD63-pHluorin plasma membrane fusion events. Each analysis pipeline comes with a set of arguments or parameters, with values that can easily be tweaked by the user in GUI of the control widget with live visual feedback of the events that are detected. The calibration of certain parameters, such as the background noise level and absolute ratiometric intensity threshold most importantly, is necessary between

experiments and between samples for optimal performance (avoiding false detections, catching all events etc), depending on the label in use and the acquisition settings such as

laser powers or exposure time of the camera. As an example of this, you can see the values of all pipeline parameters for each presented experiment in Supplementary Table 1.

3) It is a bit unclear how false negative events were calculated, or if the authors have an estimation of how many true events were missed if those images are not saved. It is referred to in Figure 2C, but this is unclear from the figure.

When it comes to the *rapid_signal_spikes* pipeline and the calcium signalling detection, see we have now added a section in the Methods that describes the procedure regarding this. False negative events were calculated from manual annotation of continuous widefield timelapses and their ratiometric, analysed counterparts (acquired without etSTED imaging and analysed post-acquisition). We find that on average, in neurons, ~75% of all events are detected, and as such leaving a false negative event ratio of ~25% on average. There is unfortunately no way of checking this directly in the etSTED experiments, as in order to optimize the temporal performance the continuous stream of widefield images are not saved, i.e. the frames where false negative events could be detected, but instead the ~10-20 widefield frames leading up to each detected event are saved, allowing to perform a check of which events are true positives and which are false positives. This check is further explained in Supplementary Fig. 6 and Supplementary Note 7.

4) It is a bit hard to work out what is going on in Fig2f and g, with the left, center, middle, and what are the arrows pointing to?

In order to increase readability, we have changed the left, center, and middle to their own subpanels, and followed the same practice for the new Figure 3 and Figure 4. The arrows in the mentioned panels are pointing to visible reorganization of the investigated structure in the STED imaging.

Decision Letter, first revision:

Dear Ilaria,

Thank you for submitting your revised manuscript "Event-triggered STED imaging" (N METH-A47452A). It has now been seen by the original referees and their comments are below. The reviewers find that the paper has improved in revision, and therefore we'll be happy in principle to publish it in Nature Methods, pending minor revisions to satisfy our editorial and formatting guidelines.

TRANSPARENT PEER REVIEW

Thank you again for your interest in Nature Methods Please do not hesitate to contact me if you have any questions.

Sincerely,
Rita

Rita Strack, Ph.D.
Senior Editor
Nature Methods

ORCID

Reviewer #1 (Remarks to the Author):

My concerns have been addressed and the manuscript is much improved. Well done!

-Henry

Final Decision Letter:

Dear Ilaria,

I am pleased to inform you that your Article, "Event-triggered STED imaging", has now been accepted for publication in *Nature Methods*. Your paper is tentatively scheduled for publication in our September print issue, and will be published online prior to that. The received and accepted dates will be Oct 6, 2021 and July 14, 2022. This note is intended to let you know what to expect from us over the next month or so, and to let you know where to address any further questions.

Please note that *Nature Methods* is a Transformative Journal (TJ). Authors may publish their research with us through the traditional subscription access route or make their paper immediately open access through payment of an article-processing charge (APC). Authors will not be required to make a final decision about access to their article until it has been accepted. Find out more about Transformative Journals

Your paper will now be copyedited to ensure that it conforms to *Nature Methods* style. Once proofs are generated, they will be sent to you electronically and you will be asked to send a corrected version within 24 hours. It is extremely important that you let us know now whether you will be difficult to

contact over the next month. If this is the case, we ask that you send us the contact information (email, phone and fax) of someone who will be able to check the proofs and deal with any last-minute problems.

If, when you receive your proof, you cannot meet the deadline, please inform us at rjsproduction@springernature.com immediately.

Once your manuscript is typeset and you have completed the appropriate grant of rights, you will receive a link to your electronic proof via email with a request to make any corrections within 48 hours. If, when you receive your proof, you cannot meet this deadline, please inform us at rjsproduction@springernature.com immediately.

Once your paper has been scheduled for online publication, the Nature press office will be in touch to confirm the details.

Once your paper has been scheduled for online publication, the Nature press office will be in touch to confirm the details.

Content is published online weekly on Mondays and Thursdays, and the embargo is set at 16:00 London time (GMT)/11:00 am US Eastern time (EST) on the day of publication. If you need to know the exact publication date or when the news embargo will be lifted, please contact our press office after you have submitted your proof corrections. Now is the time to inform your Public Relations or Press Office about your paper, as they might be interested in promoting its publication. This will allow them time to prepare an accurate and satisfactory press release. Include your manuscript tracking number NMETH-A47452B and the name of the journal, which they will need when they contact our office.

About one week before your paper is published online, we shall be distributing a press release to news organizations worldwide, which may include details of your work. We are happy for your institution or funding agency to prepare its own press release, but it must mention the embargo date and Nature Methods. Our Press Office will contact you closer to the time of publication, but if you or your Press Office have any inquiries in the meantime, please contact press@nature.com.

To assist our authors in disseminating their research to the broader community, our SharedIt initiative provides you with a unique shareable link that will allow anyone (with or without a subscription) to read the published article. Recipients of the link with a subscription will also be able to download and print

the PDF.

Nature Research journals encourage authors to share their step-by-step experimental protocols on a protocol sharing platform of their choice. Nature Research's Protocol Exchange is a free-to-use and open resource for protocols; protocols deposited in Protocol Exchange are citable and can be linked from the published article. More details can found at www.nature.com/protocolexchange/about.

Please note that you and any of your coauthors will be able to order reprints and single copies of the issue containing your article through Nature Research Group's reprint website, which is located at <http://www.nature.com/reprints/author-reprints.html>. If there are any questions about reprints please send an email to author-reprints@nature.com and someone will assist you.

Best regards,
Rita

Rita Strack, Ph.D.
Senior Editor
Nature Methods